# Otter: Generating Tests from Issues to Validate SWE Patches

**Toufique Ahmed**[1] **Jatin Ganhotra**[1] **Rangeet Pan**[1] **Avraham Shinnar**[1] **Saurabh Sinha**[1] **Martin Hirzel**[1]

## Abstract

While there has been plenty of work on generating tests from existing code, there has been limited work on generating tests from issues. A correct test must validate the code patch that resolves the issue. This paper focuses on the scenario where that code patch does not yet exist. Doing so supports two major use-cases. First, it supports TDD (test-driven development), the discipline of "test first, write code later" that has well-documented benefits for human software engineers. Second, it also validates SWE (software engineering) agents, which generate code patches for resolving issues. This paper introduces TDD-Bench-Verified, a benchmark for generating tests from issues, and Otter, an LLM-based solution for this task. Otter augments LLMs with rule-based analysis to check and repair their outputs, and introduces a novel self-reflective action planner. Experiments show Otter outperforming state-of-the-art systems for generating tests from issues, in addition to enhancing systems that generate patches from issues. We hope that Otter helps make developers more productive at resolving issues and leads to more robust, well-tested code.

## 1. Introduction

A software engineering (SWE) *issue* is a bug report or feature request for improving the code in a repository. Before a software engineer attempts to write new code that resolves a bug, they typically write a *reproduction test* to confirm the presence of the bug in the old code. In fact, even when working on a feature request, before writing new code it is recommended to create an *acceptance test* first, to confirm the presence of the feature in the new code once written. This practice is known as test-driven development (TDD), and it improves the quality of both tests and the code itself (Beck,

2002). The recent introduction of SWE-Bench (Jimenez et al., 2024) has spurred work on resolving SWE issues automatically with LLMs, typically with agents (Yang et al., 2024; Ruan et al., 2024; Wang et al., 2024b). Such issue-resolution systems are often called *SWE agents*. We posit that, just like up-front tests help human SWE engineers, they also help automated SWE agents.

While test generation is an active area of research, prior work focuses on creating tests for existing code, not on creating tests from issue descriptions alone. A recent solution SWE-Agent+ (Mündler et al., 2024) attempts to create tests from issue descriptions albeit with limited success (the state-of-the-art SWE-Agent+ only generates a fail-to-pass test in 19.2% cases and is quite expensive, requiring several LLM calls and tokens). See Section 3 for a more in-depth discussion of related work.

This paper introduces Otter (acronym for "Otter: TDD-Test gEnerator for Reproducing issues"). It takes as input $x$ an issue description and the original code in the repository, and generates as output a set of tests $y$. Otter contains a novel self-reflective action planner for deciding which code to read and which tests to write. It also uses rule-based code analyses and transformations throughout the workflow to curate LLM inputs, validate LLM outputs, and repair generated tests. Otter generates fail-to-pass tests in 31.4% of cases, and in an ensemble of size 5 dubbed Otter++, that increases to 37.0%. At the same time, the cost (even with ensembling) is less than $0.10 per issue with GPT-4o. Tests generated by Otter adhere to the same testing framework as repository's existing test suite, to which they can be added.

Today the most popular benchmark for automatically resolving SWE issues is SWE-bench Verified (Chowdhury et al., 2024). To evaluate solutions such as Otter that automatically generate tests from issues, we introduce a new benchmark, TDD-Bench-Verified. TDD-Bench-Verified evaluates tests by checking whether the tests a) fail on the old code before issue resolution, b) pass on the new code, and c) cover the code changes well. The fact that TDD-Bench-Verified is derived from SWE-bench Verified enables us to empirically study the effects of generated tests on SWE agents. We observe that the tests from Otter++ can be used to trade precision for recall on the SWE-bench Verified leaderboard. For instance, for the system ranked 3rd on the leaderboard,

[1]IBM Research, Yorktown Heights, New York, USA. Correspondence to: Toufique Ahmed <tfahmed@ibm.com>, Martin Hirzel <hirzel@us.ibm.com>.

*Proceedings of the $42^{nd}$ International Conference on Machine Learning*, Vancouver, Canada. PMLR 267, 2025. Copyright 2025 by the author(s).

filtering by the generated tests boosts precision from 60.8% to 91.9% while reducing recall to 33%. This paper makes the following contributions:

- Otter, a system that generates tests from issues, using LLMs with a novel self-reflective action planning technique along with rule-based pre- and post-processing.
- TDD-Bench-Verified[1], a benchmark for evaluating tests generated from issues, including a high-quality dataset and a metric based on test results and coverage.
- An empirical study on using tests generated from issues to filter issue-resolution candidates for SWE-bench Verified.

Generated tests can assist software engineers both before and after resolving an issue, and can increase trust in automated SWE agents.

## 2. Problem Statement

This work focuses on the problem of generating tests from issues. Specifically, the input $x$ is a pair $\langle d_{\text{issue}}, c_{\text{old}} \rangle$ of an issue description $d_{\text{issue}}$ alongside the old version $c_{\text{old}}$ of the code before the issue is resolved. The issue description is typically in natural language, but it may sometimes contain embedded code snippets or stack traces. The code is a snapshot of all the files and folders in a Python source code repository. The expected output $y$ is a set of tests that should go from failing on $c_{\text{old}}$ to passing on $c_{\text{new}}$, which is the new version of the code after the issue is resolved. By failing on $c_{\text{old}}$, the tests reproduce the issue, and by passing on $c_{\text{new}}$, they validate its resolution. Besides going from failing to passing, the tests should also maximize coverage of the code change (formalized in Section 5). A solution to this problem is thus a function $genTests$ that takes an input $x$ and returns tests $y = genTests(x)$. The new code $c_{\text{new}}$ is not available to $genTests$, which must generate tests $y$ based on $x$ alone. This is representative of the real world where source code repositories may have regression tests for existing code but lack tests for open issues. Otter provides an implementation of the $genTests$ function and, thus, a solution to this problem.

## 3. Related Work

Prior to SWE-bench (Python) (Jimenez et al., 2024), Defects4J (Java) (Just et al., 2014) has been a popular benchmark in the community. The creators of Defects4J carefully curated and cleaned up each issue by hand. Unlike SWE-bench, Defects4J only contains bug reports, no feature requests. The earliest system we are aware of that generates tests from issues, Libro (Kang et al., 2023), focuses on Defects4J. Libro achieves a fail-to-pass rate of 19.9% with one generation. (Mündler et al. (2024) ported

---

[1]https://github.com/IBM/TDD-Bench-Verified

Libro to Python and measured a fail-to-pass rate of 15.2%.) Plein et al. (2024) proposed another test-generation system for Defects4J, reporting a fail-to-pass rate of 6%. Both systems have relatively low success rates, and unlike our work, neither evaluates the impact of generated tests on issue-resolving systems.

When resolving issues, some SWE agents also generate tests along the way. The original SWE-Agent (Yang et al., 2024), a single-agent system, attempts to reproduce the issue, as explicitly instructed in its prompt. Some multi-agent systems—CodeR (Chen et al., 2024) and SpecRover (Ruan et al., 2024)—start with a Reproducer agent for generating tests. However, none of the three (SWE-Agent, CodeR, or SpecRover) are evaluated for the effectiveness of their generated tests. Agentless (Xia et al., 2024) relies on inference scaling, generating several candidate patches and several tests. It then uses the tests to help rank the patches, ultimately choosing a single patch to submit. The effectiveness of the tests is evaluated indirectly by their impact on issue resolution rate (from 27% to 32%), not directly for their own fail-to-pass rate or coverage like in our work.

Three very recent systems are dedicated to generating tests from Python issues. Aegis (Wang et al., 2024a) is a multi-agent system that uses inference scaling, but the exact dataset for their evaluation is unclear (the paper says SWE-bench Lite, but then compares against numbers from another system on SWT-bench Lite, which is different). Aegis is more costly than Otter++, and unlike our work, the Aegis paper (a) does not report coverage numbers and (b) does not evaluate how tests can help trade off precision vs. recall w.r.t. performance of SWE agents. EvoCoder (Lin et al., 2024) uses experiences from prior issues to help with the latest issue at hand, which is complementary to Otter. Unlike our work, the generated tests are not integrated with the existing CI pipeline. Furthermore, the experiments do not use execution-based metrics, making it hard to compare empirically. SWE-Agent+ adapts a patch-generating agent to generate tests instead (Mündler et al., 2024) and achieves 19.2% fail-to-pass rate on the SWT-bench Lite dataset introduced by the same paper. SWT-bench applies less rigorous quality filters than TDD-Bench-Verified, and measures coverage in a round-about way by first running additional tests than just generated ones and then subtracting them back out. Using tests from SWE-Agent+ as a filter improves precision of SWE agents to 47.8% while reducing recall to 20%. Otter++ outperforms SWE-Agent+ on all of these metrics.

CodeT (Chen et al., 2023) is one of the first approaches that leverage the same LLM to automatically generate both code samples and corresponding test cases. CodeT then executes generated code samples using the generated test cases and performs a dual execution agreement to choose the best solution and test. CodeT is not directly applicable

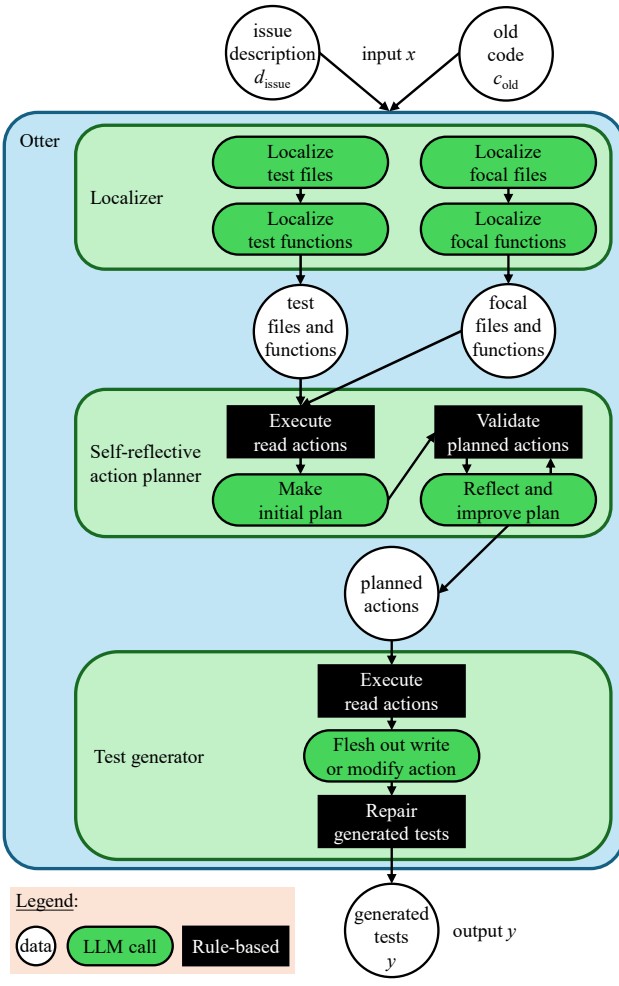

*Figure 1.* Overview of Otter.

to our current setup because we have multiple tests but no code patch. CodeMonkeys (Ehrlich et al., 2025) is another relevant work that iteratively and jointly improves a patch with a test, which is interesting and complementary to Otter. Unlike our work, their experiments do not evaluate the fail-to-pass rate nor the coverage of generated tests.

## 4. Methodology

This section describes three solutions to the problem stated in Section 2: Otter, Otter++, and a baseline approach.

### 4.1. Otter: Test Generation Guided by Self-reflective Action Planner

Figure 1 presents an overview of Otter, which has three main components: a localizer, a planner, and a test generator.

**Localizer.** Before developers tackle an issue, they usually start by gaining an understanding of the current state $c_{old}$ of

the project. They do this by localizing relevant existing tests and *focal* functions (i.e., functions exercised by those tests and likely places for a fix). Depending on the developer's familiarity with the project and the nature of the issue, the difficulty of localization may vary. Inspired by human developers' actions, our approach also starts with test and focal function localization. The localizer phase collects all files from $c_{old}$ that contain at least one test function for test localization. It presents the list of files and the issue description $d_{issue}$ to the LLM and asks it to generate 10 relevant test file names. Our initial findings on the dev set of SWE-bench indicated that the Top-10, Top-5, and Top-1 accuracy for test file localization with GPT-4o are 83.6%, 76.0%, and 59.1%, respectively. We restricted file retrieval to 10 files so as not to overwhelm the contexts for subsequent LLM calls. Next, the localizer validates the file names by comparing them with the previously collected file list and drops the ones that do not match. After localizing test files, it makes a second LLM call with file names and test function names from those files. The model chooses the test files and functions relevant to $d_{issue}$. The localizer again validates the retrieved file names, but instead of dropping any hallucinated file names, replaces them with the file names with minimal edit distance. This ensures validity of identified files, which is essential for the subsequent test-generation phase. Note that even if the localizer chooses the wrong test file, if the file is at least valid, the test generator may still succeed. Focal file and function localization follows a similar two-LLM-call approach. Figures 14 and 15 in the Appendix illustrate the LLM prompts for localization.

**Self-Reflective Action Planner.** The second phase of Otter creates a *plan*, which is a list of actions for generating the fail-to-pass tests $y$. There are three kinds of actions: read, write, or modify. A *read f* action reads a function $f$ from $c_{old}$ to use as context in a prompt. A *write f* or *modify f* action declares the intent to write a new test function or modify an existing test function. Here, $f$ is a file and function name (in the planner, write and modify actions do not yet include the exact code for the test function, which is left to the test generator phase of Otter). The planner starts by executing read actions for the files and functions provided by the localizer. Next, it prompts an LLM with the function definitions, the issue description $d_{issue}$, and instructions to make an initial plan, restricted to only read actions. The next step of the planner validates the planned actions: it checks whether the actions generated by the LLM refer to valid file and function names. The final step of the planner is "reflect and improve plan", an LLM call with a prompt including feedback from validation. At this point, the plan is no longer restricted to only read actions, and can also contain a write or modify action. Section B in the appendix provides statistics on read/write/modify actions generated by the planner. The model is also instructed to self-reflect on

the proposed plan with one of three possible outcomes: "Satisfied", "Unsatisfied", and "Unsure". If the model chooses "Satisfied", Otter moves forward to the test-generation phase. For other options, it returns to the validation step and then repeats the "reflect and improve plan" step. This process is repeated at most five times. In most cases, the model is satisfied with the plan in the first two turns. The two planner prompts are presented in Figures 18 and 19 (Appendix).

**Test Generator.** The test-generation phase of Otter executes the actions computed by the planner, which can involve generating a new test (write action) or updating an existing test (modify action). To guide the LLM in this task, we extract the test structure and imports from the localized test file and make them available in the prompt. This reduces the burden on the LLM to generate imports. The file structure is relevant for new tests, to determine their insertion point in the test file. Otter uses a different prompt for a write vs. a modify action, illustrated in Figures 20 and 21 in the Appendix. For new tests, the model needs to generate the preceding function name in addition to the test. Unlike most SWE agents, Otter does not try to generate diffs; instead, it asks the model to generate complete test functions, even for the modification case. Since model training data tends to contain more complete functions than diffs, we expect the model to perform better at generating functions.

To handle missing or hallucinated imports, Otter includes an import-fixing step in this phase, where it looks at model-generated imports and linting errors detected using Flake8[2] (a static analysis tool) to identify missing imports. Note that Flake8 reports different styling errors; we manually curated the error codes to catch name-related errors. In case of missing imports, we add a dummy import to the function. Then, we take the model-generated and dummy imports and try to find the imported module among the files in the codebase. If we find the module, we replace the model-generated/dummy import with the one from the codebase; otherwise, we continue with the model-generated/dummy import as a fallback. Finally, we add the function in the codebase and generate a git diff to create the test patch.

## 4.2. Otter++: Ensemble using Multi-Sampling with Heterogeneous Prompting

Otter has several components (e.g., localizer, planner), but they are not perfect. The file localizer accuracy for focal and test localization is 82.4% and 70.6% with GPT-4o, respectively. As the output from localization serves as input to later LLM calls, those later calls may be affected by inaccuracies in localization. Conversely, LLMs can sometimes generate fail-to-pass tests in a zero-shot setup, even without any context. So, selectively including and excluding parts

of localization may produce different fail-to-pass tests that are not generated by Otter. Otter++ uses the test generated by Otter (T1) and adds four new tests (T2–T5), obtained by skipping the planner stage, and including neither, one, or both of focal and test localization. In other words, Otter++ runs the test generator stage five times with different, *heterogeneous* prompts. We favor heterogeneous prompts with greedy decoding over homogeneous prompts with higher model temperature because our initial experiments showed that the latter yields poorer tests and lower diversity.

To pick a single test among the five candidate tests, we run the five tests on $c_{old}$ (recall that $c_{new}$ is not available yet) and analyze the execution logs. If a test passes on $c_{old}$, we just discard that test because it violates the fail-to-pass criterion. We classify the remaining tests into three groups: assertion failure, other failure (the test runs but produces the wrong output), and error (the test does not run properly, e.g., because of wrong syntax or an exception in a fixture). We pick a test from the first non-empty group to maximize the chance that it failed for the right reason (i.e., it reproduces the bug described in the issue). If the selected group has multiple tests, we break the tie with a pre-determined ordering of the five prompts that favors tests from prompts with more or better information: Otter, followed by the prompts with both localizers, test localizer only, focal localizer only, and neither localizer, in that order.

### 4.3. Baseline: Zero-shot Test File Generation

Recent instruction-tuned LLMs excel at following instructions (Peng et al., 2023; Zhang et al., 2023). We propose a simple zero-shot approach to generate a fail-to-pass test given the repository name and the issue description. Given the prompt, the model generates a complete test file with all necessary imports to make it compilable. In real scenarios, test files usually have multiple test cases, but this baseline usually generates only a single test per file.

## 5. TDD-Bench-Verified Benchmark

This section presents TDD-Bench-Verified, a new benchmark that supports evaluation of techniques for generating tests from an issue description and an old code version, without access to new code that resolves the issue (see problem statement in Section 2).

### 5.1. TDD-Bench-Verified Evaluation Harness

Figure 3 shows the harness for evaluating tests $y$, which typically come from a *genTests* solution, but the harness can also be applied on golden tests $\hat{y}$ mined from a pull request (PR). The evaluation harness runs in a containerized environment. Starting at the top left, tests come in the form of a patch, which is applied (via `git apply`) on the old

---

[2]https://flake8.pycqa.org/en/latest/

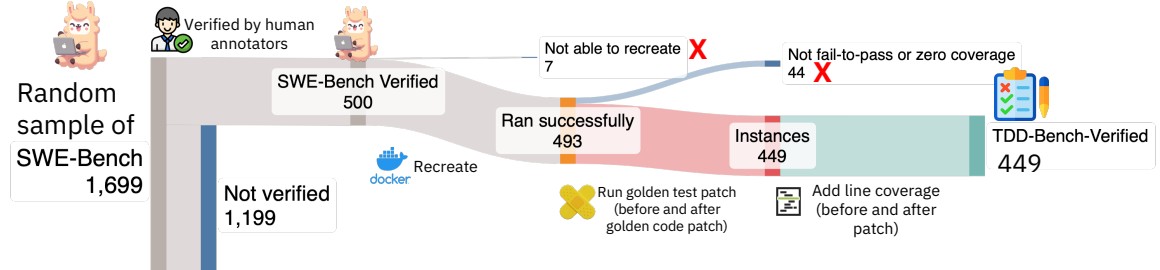

*Figure 2.* Overall flow of TDD-bench dataset filtering starting from SWE-bench verified.

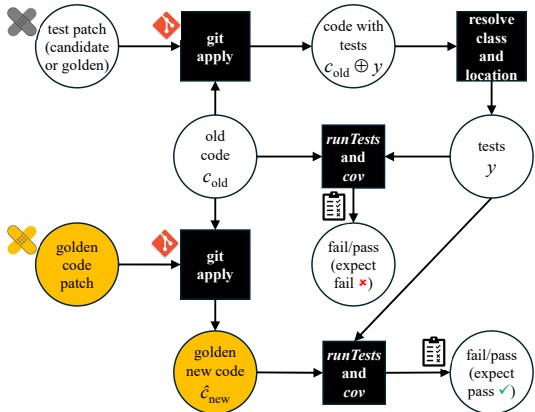

*Figure 3.* Evaluation harness for TDD-Bench-Verified.

code $c_{old}$. Next, the harness analyzes the resulting code $c_{old} \oplus y$ to resolve the contributed test functions $y$. The harness then executes $y$ while avoiding running other tests that occur in the same file but were not contributed in the patch. This yields test results, including coverage achieved on the old code. At least one of these tests should fail, indicating that the tests reproduce the issue at hand.

Moving to the bottom part of Figure 3, the code changes come from the golden code patch mined from the same PR, which is applied on $c_{old}$ to obtain the golden new code $\hat{c}_{new}$. The harness executes the tests $y$ again, this time on the new code, to obtain a second set of test results. This time, all tests should pass, to validate that the issue was indeed resolved. An example test patch is presented in Figure 9.

### 5.2. Dataset Filters

TDD-Bench-Verified builds upon prior work from SWE-bench (Jimenez et al., 2024) and SWE-Bench Verified (Chowdhury et al., 2024). SWE-bench uses filters to keep only those mined instances $x$ for which the set of golden tests $\hat{y}$ contains at least some tests that fail on $c_{old}$ and pass on the golden new code $\hat{c}_{new}$ from the same PR. SWE-Bench Verified is a subset of SWE-Bench, consisting of 500 instances further vetted by human annotators (Chowdhury et al., 2024). The annotators filtered out instances where the issue description $d_{issue}$ was underspec-

ified or the golden tests $\hat{y}$ were overly specific, i.e., would reject some valid new code $c_{new}$. They also removed some instances where tests failed due to environment problems.

In the same spirit, TDD-Bench-Verified applies more filters to obtain an even higher-quality dataset. In a nutshell, the filtering process applies the TDD-Bench-Verified evaluation harness (Figure 3) to the golden tests $\hat{y}$ from the original PR. Specifically, substituting $\hat{y}$ wherever $y$ occurs in Figure 3 checks whether the PR indeed contributed tests that went from failing to passing. We filter out any instance where the contributed tests do not satisfy that criterion. Although the human annotators of SWE-bench Verified were diligent, a few instances slipped past their filters, and we drop those for TDD-Bench-Verified.

Figure 2 illustrates the filtering process. Starting from the 500 instances of SWE-bench Verified, we first drop 7 instances whose environment we could not recreate. Next, we run the test harness on the golden tests $\hat{y}$. This filters out 44 additional instances because the tests do not have the expected fail-to-pass behavior (25 instances) or have zero line coverage on the golden code patch (19 instances). In the end, 449 high-quality instances remain across 12 repositories. Table 7 summarizes key statistics of TDD-Bench-Verified.

### 5.3. Evaluation Metric

Passing a test does not necessarily mean a patch is adequate to address the issue. Aleithan et al. (2024) reported that 31.1% of the passed code patches in SWE-Bench are suspicious due to weak test cases. To evaluate test adequacy, we compute the coverage of the submitted test patch. One key difference between SWE-Bench Verified and TDD-Bench-Verified is that the former runs an entire test file to evaluate the submitted patch, whereas we only run the contributed tests $y$. Not running other test cases enables us to precisely compute coverage of $y$. If the tests are relevant, they should cover the deleted lines in $c_{old}$ and the added lines in $\hat{c}_{new}$. We integrated the Python Coverage package into the 12 repositories and updated the test scripts to allow us to run specific test cases and compute coverage for them.

We define the $tddScore$ metric that evaluates the quality of tests generated by a solution $genTests$ over a set

$X = \{x_0, x_1, \ldots\}$ of instances. It returns a number between 0 and 100, the higher the better. It is defined as 100 times the arithmetic mean of the per-instance scores:

$$tddScore(X, genTests) = \frac{100}{|X|} \sum_{x \in X} tddScore\big(x, genTests(x)\big)$$

Given a set of tests $y = genTests(x)$ submitted for an instance, the per-instance score is a product of two factors:

$$tddScore(x, y) = failToPass(x, y) \cdot adequacy(x, y)$$

The first factor is a binary correctness metric, using the indicator function for the tests $y$ failing on the old code times the indicator function for the tests $y$ passing on the new code. While the solution $genTests$ only has access to the old code $c_{\text{old}}$, the evaluation metric also uses the hidden golden new code $\hat{c}_{\text{new}}$ right after the issue was fixed.

$$failToPass(x, y) = \\ I\big(fail \in runTests(y, c_{\text{old}})\big) \cdot I\big(fail \notin runTests(y, \hat{c}_{\text{new}})\big)$$

The second factor is a fraction between 0 and 1 based on test coverage on the old and new code:

$$adequacy(x, y) = \\ \frac{|cov(y, c_{\text{old}}) \cap (c_{\text{old}} \setminus \hat{c}_{\text{new}})| + |cov(y, \hat{c}_{\text{new}}) \cap (\hat{c}_{\text{new}} \setminus c_{\text{old}})|}{|c_{\text{old}} \setminus \hat{c}_{\text{new}}| \quad + \quad |\hat{c}_{\text{new}} \setminus c_{\text{old}}|}$$

Adequacy focuses on just the coverage of lines added and deleted when going from the old code to the new code, because those are the most relevant lines to be tested. In the above, $cov(y, c)$ is the set of lines covered by running tests $y$ on code $c$; $(c_{\text{old}} \setminus \hat{c}_{\text{new}})$ is the set of lines deleted by the PR patch; and $(\hat{c}_{\text{new}} \setminus c_{\text{old}})$ is the set of lines added by the PR patch. We evaluate adequacy jointly for added and deleted lines, as some code patches may contain only added or deleted lines.

# 6. Evaluation

We conducted an extensive set of empirical studies, evaluating the effectiveness of our approach (§6.2) and the components of Otter and Otter++ (§6.3), comparing our approach against existing techniques (§6.4), and investigating the cost effectiveness of Otter (§6.5), characteristics of the generated tests (§6.6), usefulness of the generated tests in supporting automated program repair (§6.7), and possible effects of data contamination (§6.8).

## 6.1. Experiment Setup

The evaluation used the closed-source GPT-4o (gpt-4o-2024-08-06) and the open-source Mistral-large model (123 billion parameters). All experiments used greedy decoding. For each instance, Otter makes 7–11 LLM calls for T1. Otter++ makes one additional call for each of the other four tests (T2-T5) after the localization stage. To evaluate using the generated tests for SWE agents, we conducted a large-scale

*Table 1.* Performance of Otter, Otter++, and baseline technique on TDD-Bench-Verified.

| Model | Approach | # of fail-to-pass test | # of fail-to-pass test in (%) | tddScore | Coverage |
|---|---|---|---|---|---|
| Mistral-Large | Zero-shot | 57 | 12.7 | 11.8 | 60.6 |
| | Otter | 121 | 26.9 | 25.1 | 70.5 |
| | Otter++ | 144 | 32.1 | 28.6 | 70.4 |
| GPT4o | Zero-shot | 84 | 18.7 | 17.2 | 60.0 |
| | Otter | 141 | 31.4 | 29.4 | 70.6 |
| | Otter++ | 166 | 37.0 | 32.4 | 71.5 |

*Table 2.* Contribution of each component of Otter.

| Model | Component | Approach | # of fail-to-pass | tddScore | Change in tddScore% |
|---|---|---|---|---|---|
| Mistral-large | - | Otter (T1) | 121 | 25.7 | - |
| | Action Planner | without Action Planning (complete)* (T2) | 96 | 20.2 | -21.4 |
| | | without Plan Refinement (Just 1 attempt) | 115 | 23.8 | -7.4 |
| | | without Action Validation | 107 | 22.1 | -14.0 |
| | Localizers | without Focal Localization* (T3) | 96 | 20.2 | -21.4 |
| | | without Test Localization* (T4) | 77 | 16.5 | -35.8 |
| | | without Focal & Test Localization* (T5) | 81 | 17.2 | -33.1 |
| | Test Generator | without Fixing Import | 117 | 24.6 | -4.3 |
| | | without Imports at Generation | 114 | 23.7 | -7.8 |
| GPT-4o | - | Otter (T1) | 141 | 29.4 | - |
| | Action Planner | without Action Planning (complete)* (T2) | 110 | 23.6 | -19.7 |
| | | without Plan Refinement (Just 1 attempt) | 130 | 27.5 | -6.5 |
| | | without Action Validation | 120 | 25.7 | -12.6 |
| | Localizers | without Focal Localization* (T3) | 115 | 25.2 | -14.3 |
| | | without Test Localization* (T4) | 107 | 24.2 | -17.7 |
| | | without Focal & Test Localization* (T5) | 110 | 24.2 | -17.7 |
| | Test Generator | without Fixing Import | 128 | 26.7 | -9.2 |
| | | without Imports at Generation | 130 | 26.7 | -9.2 |

* is not followed by action planning

experiment with 22 systems from the SWE-Bench leaderboard. We ran $22 \times 449 \times 5 = 49{,}390$ Docker containers or tests (one Docker container per test) to report the results.

## 6.2. Effectiveness of Otter, Otter++, and the Baseline

Table 1 shows that GPT-4o-based Otter and Otter++ perform well on test generation, creating fail-to-pass tests for 31.4% and 37% of the instances, respectively, whereas the baseline produced such tests for 18.7% of the instances. The improvements are also reflected in $tddScore$. We observe similar performance improvements with Mistral-large. The pass@5 rate (where one of the five tests is fail-to-pass) for Otter++ is 44% for GPT-4o and 37% for Mistral-large.

## 6.3. Ablation Study

Table 2 shows the ablation study for Otter (T1). We also present the individual performance for the other four tests (T2–T5) produced by Otter++. We can see that all the components contribute to Otter's performance. Without action planning, we lose more than 14%–20% of fail-to-pass tests for GPT-4o and 21%–36% of the tests for Mistral-large.

## 6.4. Comparison with the Approaches of Mündler *et al.*

Mündler et al. (2024) proposed a set of approaches for generating fail-to-pass tests. We ran Otter and Otter++ on their

*Table 3.* Comparing with approaches proposed by Mündler et al. (2024) on the 276 instances of their SWT-Lite.

| Approach | # of Fail-to-pass Tests | in (%) |
|---|---|---|
| ZeroShot (GPT-4) | 16 | 5.8 |
| ZeroShotPlus* (GPT-4) | 28 | 10.1 |
| LIBRO* (GPT-4) | 42 | 15.2 |
| AutoCodeRover (GPT-4) | 25 | 9.1 |
| SWE-Agent (GPT-4) | 46 | 16.7 |
| SWE-Agent+ (GPT-4) | 53 | 19.2 |
| **Otter** (GPT-4o) | **70** | **25.4** |
| **Otter++** (GPT-4o) | **80** | **29.0** |

\* uses "proposed patch" while generating tests

dataset to study how the approaches compare. They also evaluated zero-shot approaches, which differ from our zero-shot baseline. All of their approaches (including the zero-shot ones) instruct the model to generate a novel code diff format introduced by their paper. Two of their approaches use a proposed patch in the prompt. Mündler *et al.*'s SWE-agent and SWE-agent+ approaches are derived from SWE-Agent, which was originally designed for generating code patches (Yang et al., 2024). Table 3 shows the results. Otter and Otter++ perform better than their best-performing approach, generating 70 (25.4%) and 80 (29.0%) fail-to-pass tests compared to 53 (19.2%) fail-to-pass tests generated by SWE-agent+.

Otter++ uses execution logs on the current code base $c_{old}$ which give it an advantage in the final selection stage. The feedback works as a contributing factor to the superior performance of Otter++. Note that Otter does not use any execution logs, yet it still performs significantly better than LIBRO (25.4% vs. 15.2% in Table 3). The novel code diff format proposed by Mündler *et al.* has similarities to our approach. However, their format requires the model to perform additional tasks such as writing the file name, change type, and line number in response to one LLM call. Additionally, to instruct the model to generate a specific format, the authors had to include an example that is not relevant to the issue itself. In our generation step, the model only needs to generate the test (and prior function name for positioning new tests).

### 6.5. Cost Effectiveness of Otter

Table 4 presents the cost for invoking the GPT-4o model to process 449 instances with Otter. Each sample requires an average of $0.06. The "reflect and improve plan" step can make 1–5 LLM calls. Figure 4 shows that, for more than 80% of the instances, both GPT-4o and Mistral are satisfied within two calls. Therefore, the total calls vary from 7–8 for most instances. The cost is very low because we do not accumulate context from prior calls in subsequent calls. We do not discuss the cost for Mistral-large because the

*Table 4.* Cost for running Otter and Otter++ with GPT-4o.

| Component | Cost | Cost/Sample |
|---|---|---|
| Focal Localization | $8.61 | $0.02 |
| Test Localization | $9.63 | $0.02 |
| Action + Generate | $10.94 | $0.02 |
| **Total for Otter** | **$29.18** | **$0.06** |
| Additional Tests (T2-T5) | $11.20 | $0.02 |
| **Total for Otter++** | **$40.38** | **$0.09** |

*Table 5.* Comparing the coverage of model-generated and developer-written golden tests.

| Model | Is Fail-to-pass? | Otter | Golden test |
|---|---|---|---|
| Mistral-large | Yes | 93.1 | 95.5 |
| | No | 63.8 | 93.4 |
| GPT-4o | Yes | 93.7 | 95.6 |
| | No | 60.0 | 93.3 |

model was hosted locally. Otter++ makes four additional calls and reuses the output from localizers. The total cost for Otter++ (which includes Otter) is $0.09 per instance.

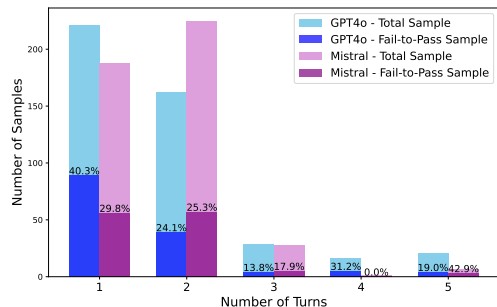

*Figure 4.* Number of turns taken in "reflect and improve plan" step.

### 6.6. Characteristics of the Generated Tests

**Coverage of the generated tests.** We compared the coverage achieved by the Otter-generates tests and the golden tests written by developers. We observe that, for the fail-to-pass tests, Otter-generated and golden tests have very similar and high coverage—more than 90% with both models (Table 5). However, for the other tests, the coverage is quite low for Otter. This indicates that tests with higher coverage are more likely to be fail-to-pass tests.

**Prompts complementarity.** Figure 5 presents the overlap among instances for which fail-to-pass tests could be generated by different prompts using the GPT-4 model. Overall, each of the prompts is successful on some instances on which the other prompts fail, with T1 (Otter with GPT-4o) achieving the most success in this respect—producing fail-to-pass tests for 24 instances on which none of the other prompts succeeded in generating such tests. Thus, combining the results from the different prompts increases the fail-to-pass rate to 44% (38% for Mistral) at pass@5. This

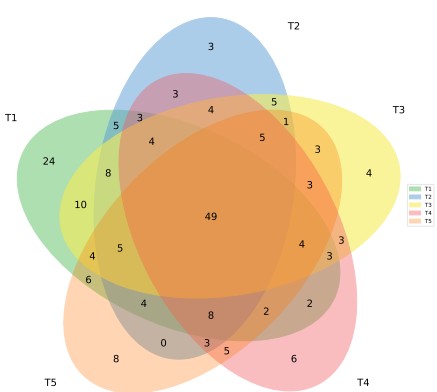

*Figure 5.* Number of instances with fail-to-pass tests generated by different prompts.

*Table 6.* Analysis of the Otter-generated tests using GPT-4o model from different perspectives.

| Perspective | Category | # of Sample | #of fail-to-pass | fail-to-pass rate |
|---|---|---|---|---|
| Type of Test | PatchExisting | 122 | 50 | 41.0 |
| | AdditionOnly | 327 | 91 | 27.8 |
| Test on old Codebase | Pass | 69 | 0 | 0 |
| | AssertionFail | 170 | 97 | 57.1 |
| | Fail | 101 | 40 | 39.6 |
| | Error | 109 | 4 | 3.7 |
| Focal Localization | Correct | 370 | 118 | 31.9 |
| | Wrong | 79 | 23 | 29.1 |
| Test Localization | Correct | 317 | 117 | 36.9 |
| | Wrong | 132 | 24 | 18.2 |

indicates the potential of incorporating different prompts for test generation.

**Detailed analysis of the tests.** Table 6 presents the analysis of the Otter-generated tests using GPT-4o model from different perspectives. We see that the fail-to-pass rate is higher when patching an existing test than when adding a new test, which is expected because the model can have better context from the existing test. Writing a new test is inherently more difficult than modifying an existing test. It is expected that the test will fail on the old codebase. However, we found 69 samples in Otter where the test passed on the old codebase. Also, our analysis shows that tests with assertion failures have a higher success rate (57.1% fail-to-pass rate) compared to other groups. We did not see much impact of focal localization on the performance, with 31.9% and 29.1% for correct and incorrect localization, respectively. Test localization has a significant impact on the performance (36.9% vs. 18.2% for correct and incorrect localization). In our ablation study, we also found that test localization is more important than focal localization. In Appendix E, we discuss the impact of hallucination replacer in our pipeline.

**Heterogenous prompts vs. temperature.** Otter++ scaled well with samples up to 5, giving 0.5%-5.6% improvement (see Appendix C). Otter++ uses heterogeneous prompting

instead of higher temperature to generate multiple samples. We make multiple LLM calls in different stages and multi-sampling in each stage would exponentially increase the test counts. Therefore, to compare heterogenous prompting with temperature, we generated 5 samples at high temperature (1.0) in the last LLM call in the Test Generator phase of Otter (our best solution). Though the average number of fail-to-pass test goes up (117.8 vs. 116.6), the Top-5 and Top-1 (using Otter++'s ranker) results remain lower (Top-5: 173 vs. 197 and Top-1: 146 vs. 166). That means heterogenous prompting boosts up the overall ensemble performance if we compare at the same number of samples. Appendix D discusses more details.

### 6.7. Test Generation and SWE Agents

The tests generated by our approach can be used for filtering bad code patches and increasing the precision of solutions proposed by different systems from the SWE-bench Verified leaderboard. We take the top 22 systems from the leaderboard and run the five tests generated by Otter++. We filter out a code patch if all the tests fail on it. Figure 6 shows that this achieves a precision of 65% to 92% while maintaining a decent recall of 30%-41%, except for one system where the precision increased by 22% to 167%. Note that Mündler et al. (2024) achieved 47.8% precision at 20% recall on SWE-Agent. Using tests generated by Otter++ achieved much higher precision while maintaining greater recall.

Apart from filtering SWE-patches, we could use our tests to choose the best SWE-patch, which would be a good application of the Otter-generated tests. We have tried CodeT ranking on candidates from the top 3 leaderboard systems on SWE-Bench-Verified and observed 2% improvement. Note that some of these leaderboard solutions have already been through good rankers and used superior models. Improving upon these samples using a ranker may be difficult.

### 6.8. Effect of Data Contamination

As TDD-Bench-Verified is based on historic GitHub issues, they may be included in the pre-training data of the LLMs we use. To see whether the model simply generates memorizes tests, we performed two different experiments.

**Model data cut-off date.** The cutoff date for the GPT-4o model is October 2023. Unfortunately, we have only one sample dated post-cutoff and could not compare the two groups. Popular GitHub repositories evolve quickly. It is likely that only the snapshot taken during the data collection process was seen by the model and that it performed well on a specific year of data. Figure 7 shows the total and fail-to-pass test distribution by year. We did not observe any pattern in performance among the distributions by year. For example, for 2020, 2021, and 2023, we have very similar

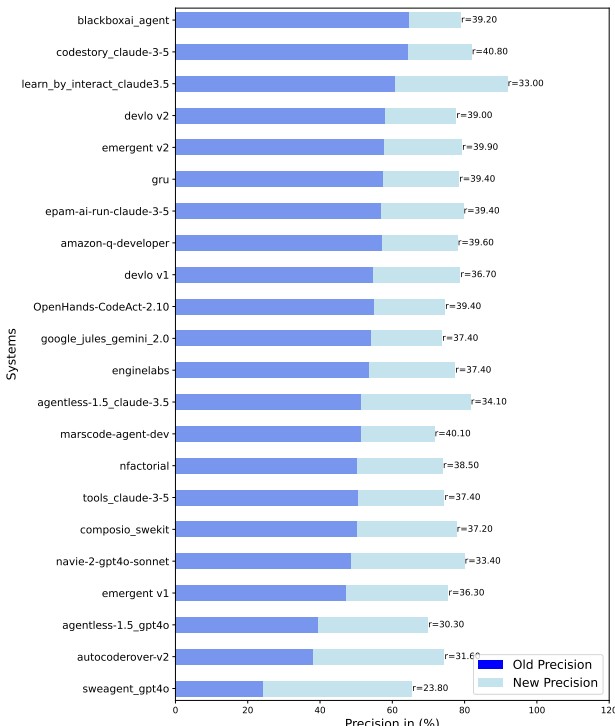

*Figure 6.* Precision of all 22 systems from the leaderboard. Recall is also mentioned at the top of each bar.

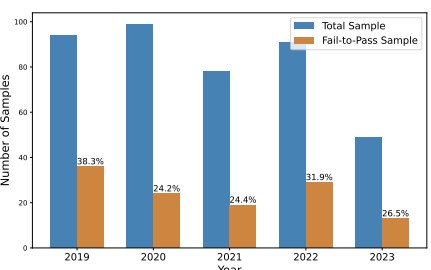

*Figure 7.* Sample distribution of total and fail-to-pass tests by year.

performance by the model.

**Test similarity.** We conducted another experiment following the approach of Schäfer et al. (2024). We compute the similarity score between the generated test from Otter and the most similar test from the repository, as follows: $max_{tp \in TP}(1 - \frac{dis(t^*, t_p)}{max(len(t^*), len(t_p))})$, where $TP$ is the set of test functions and $t^*$ is the generated test. Figure 8 shows the similarity scores for new tests. For 90% of the instances, the similarity score is less than 0.6. Table 12 in the appendix shows some samples with more than 0.5 similarity to give the reader some idea. From our observation, even at similarity score of 0.7, the tests are significantly different. As expected, for modified tests, the similarity is higher. However, we did not observe any difference between fail-to-pass tests and other tests (see appendix for figures). Therefore, the model is not simply generating memorized tests.

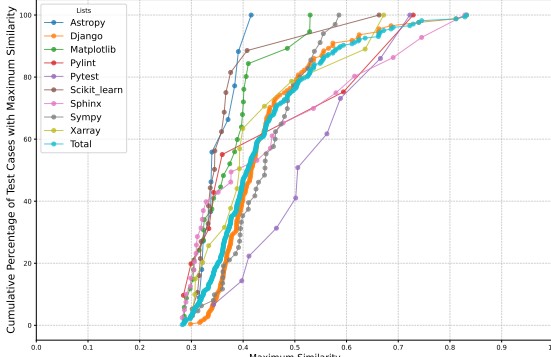

*Figure 8.* Cumulative percentage of Otter-generated new tests, using GPT-4o, with maximum similarity less than the similarity value shown on the x-axis.

## 7. Limitations

One limitation of TDD-Bench-Verified is that it is mined from 12 popular Python repositories, so our findings may not apply to other programming languages and repositories. We note that SWE-bench, despite having the same limitation, has been impactful, and one of the findings in the SWE-bench paper is that "difficulty does not correlate with issue resolution date", indicating that contamination problems (if any) are minor (Jimenez et al., 2024). Our results on data contamination (§6.8) indicate the same thing. A limitation of Otter is that it considers only one test file and generates only one block of code. In real-world projects, test code can be spread across multiple files or blocks of code. Despite that, Otter exceeded the state-of-the-art performance, so we leave further improvements to future work. We use the Python `coverage` package for computing test coverage, but this package can fail for various reasons, such as permission issues, version incompatibility, or configuration problems. In Otter, we computed coverage for all projects, including SymPy. However, upon manual validation, we found the coverage information for SymPy to be unreliable. Therefore, we removed coverage from the final $tddScore$ metric for SymPy instances (<15% of the total instances). Note that coverage does not affect the reported fail-to-pass scores.

## 8. Conclusion

The primary contribution of this paper is Otter, a system for generating tests from issue descriptions before issue resolution. Otter outperforms the prior state of the art in fail-to-pass tests generated while also costing less. This paper also contributes TDD-Bench-Verified, a new benchmark for the same problem statement, mined from real-world GitHub issues with strict filters and evaluation metrics. Finally, this paper demonstrates that generated tests can improve patches generated by SWE agents, helping them reach a precision of between 65% and 92%. TDD-Bench-Verified and Otter generated tests are at https://github.com/IBM/TDD-Bench-Verified.

## Impact Statement

This paper presents work whose goal is to advance the field of software testing and program repair. Developers spend a significant amount of time resolving bugs and testing them. We believe this work will significantly improve the developers' experience in their day-to-day life. We envision two ways of integrating Otter into existing developer workflows. First, Otter can run on a new issue to propose a test, which the stakeholders can use to clarify requirements for the desired behavior after issue resolution and, following that, a developer can use the test for test-driven development to resolve the issue. Second, Otter can be paired with a patch-generation solution (a "SWE agent") to create a PR that includes both a patch and a test.

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

## A. Different attributes of the TDD-Bench-Verified Instances & Sample Test Patch

Table 7 presents different attributes of the TDD instances to give some idea about the nature of the issues we are dealing with in TDD-Bench-Verified. We also present a sample test patch in Figure 9.

*Table 7.* Different attributes of the TDD-Bench-Verified instances.

| Project | # of Instances | Fraction of Dataset (in %) | # of Files | # of Test Files | Average # of Lines Deleted and Added | | Average Word Count in Issue Description |
| --- | --- | --- | --- | --- | --- | --- | --- |
| | | | | | On Code | On Tests | |
| Astropy | 18 | 4.0 | 1,990 | 351 | 11.9 | 28.7 | 304.5 |
| Django | 212 | 47.2 | 6,863 | 810 | 12.0 | 24.7 | 145.6 |
| Flask | 1 | 0.2 | 275 | 27 | 3.0 | 5.0 | 35.0 |
| Matplotlib | 32 | 7.1 | 4,656 | 102 | 9.3 | 20.0 | 260.5 |
| Pylint | 10 | 2.2 | 3,833 | 51 | 24.7 | 33.8 | 347.1 |
| Pytest | 16 | 3.6 | 639 | 114 | 24.6 | 53.5 | 250.1 |
| Requests | 5 | 1.1 | 155 | 9 | 3.6 | 6.6 | 85.2 |
| Scikit-learn | 25 | 5.6 | 1,772 | 242 | 11.8 | 17.1 | 297.6 |
| Seaborn | 2 | 0.5 | 353 | 34 | 13.5 | 18.5 | 182.5 |
| Sphinx | 41 | 9.1 | 1,917 | 137 | 17.5 | 26.1 | 186.2 |
| Sympy | 67 | 14.9 | 2,050 | 617 | 12.1 | 11.9 | 114.2 |
| Xarray | 20 | 4.5 | 394 | 67 | 17.1 | 24.3 | 301.0 |
| Overall | 449 | 100.0 | 24,897 | 2,561 | 13.2 | 23.3 | 182.0 |

*File counts are based on the main branches of the project (cloned on October 29, 2024).

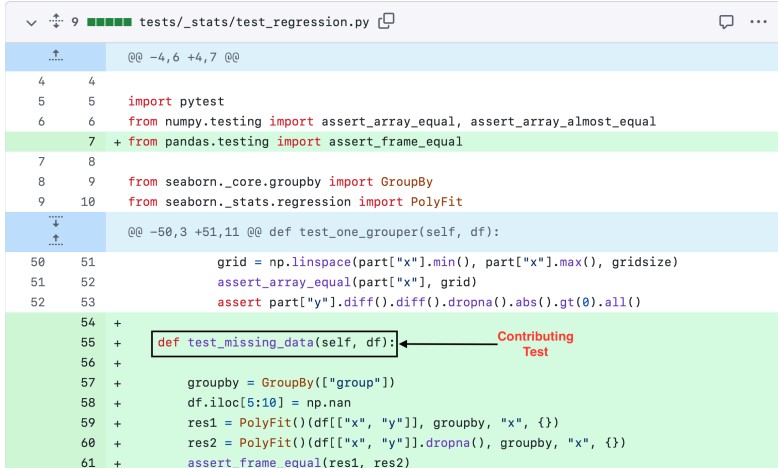

*Figure 9.* Example test patch with one contributed test. Although the test file name `test_regression.py` and test name `test_missing_data` are available in this diff, the class `TestPolyFit` enclosing `test_missing_data` is missing. By applying the test patch to the base commit and parsing the file, we retrieve `TestPolyFit`, which is required to run `test_missing_data`.

## B. Action Counts in Self-Reflective Action Planner Phase

Table 8 presents the stats on the different types of actions proposed by the model in the self-reflective action planner phase.

## C. Scaling of Otter++

Otter++ uses heterogeneous prompting and execution logs to select the best solution. Therefore, we cannot significantly increase the number of samples. Otter++ scaled well with samples up to 5, giving 0.5%-5.6% improvement (see Table 9).

*Table 8.* Stats on action count.

| Model | Action | Average | Max | Min |
|---|---|---|---|---|
| GPT-4o | Read | 5.4 | 21 | 3 |
| | Write | 1.2 | 6 | 0 |
| | Modify | 0.5 | 5 | 0 |
| Mistral-large | Read | 6.5 | 20 | 2 |
| | Write | 1.2 | 5 | 0 |
| | Modify | 0.5 | 5 | 0 |

*Table 9.* Scaling of Otter++

| # of Candidate | # of Fail-to-pass | In % |
|---|---|---|
| 1 | 141 | 31.4 |
| 2 | 154 | 34.3 |
| 3 | 157 | 35.0 |
| 4 | 164 | 36.5 |
| 5 | 166 | 37.0 |

## D. Heterogenous Prompts vs Temperature

In Table 10, the Top-1 column shows "fail-to-pass @ 1" based on our actual ranker, whereas the Oracle column shows "fail-to-pass @ N", i.e., the result if we had a perfect ranker. Heterogeneous prompts yield better results than high-temperature samples in both the Top-1 column and the Oracle column. This is evidence that the difference indeed comes from the diversity of heterogeneous prompts and is not ranker specific. The difference between the Top-1 and the Oracle column is almost the same in the first three rows (i.e., all settings with 5 samples). That said, there is still room for improvement in the ranker, as illustrated by the last row. In the last row, an oracle that could reliably choose among all 10 samples would perform amazingly well, motivating further work on better rankers.

## E. Impact of hallucination replacer.

The localizer makes two LLM calls during which hallucinations can occur. In the first call, we drop hallucinated file names. In the second call, we replace hallucinated file names with existing ones to ensure our pipeline functions correctly. Table 11 show that our proposed technique helps Otter achieve better performance. For example, if we didn't replace the hallucinated file name in the 2nd call of test localization (the last row), we would have lost 5 samples because our pipeline would have exited with an error. Replacing the hallucinated name generated 4 fail-to-pass tests for those samples.

## F. Data Contamination

Figure 10 to Figure 13 present cumulative percent of Otter generated test with different criteria, using GPT-4o, with maximum similarity less than the similarity value shown on the x-axis. Table 12 shows examples with higher ($> 0.5$) similarity score.

## G. Prompts for Otter

Figure 14 to Figure 21 present all the prompts used for Otter.

*Table 10.* Performance of Our Test Generation Approach in Different Settings.

| Setting | Individual Performance | Max | Average | Top-1 | Oracle |
|---|---|---|---|---|---|
| 5 heterogeneous prompts (Otter++) | 141, 110,115, 107, 110 | 141 | 116.6 | 166 | 197 |
| 5 high-temperature samples (temp=1.0) | 111, 118, 126, 121, 113 | 126 | 117.8 | 146 | 173 |
| 5 = 1 greedy + 4 high-temperature samples | 141, 111, 118, 126, 121 | 141 | 123.4 | 152 | 183 |
| 10 = 5 heterogeneous + 5 high-temperature | 141, 110,115, 107, 110, 111, 118, 126, 121, 113 | 141 | 117.2 | 168 | 218 |

*Table 11.* Impact of hallucination replacer on GPT-4o generated tests.

| Focal/Test | LLM Call | #of Halluciantion Happen | #of fail-to-pass | #of fail-to-pass rate |
|---|---|---|---|---|
| Focal | LLM Call 1 | 19 | 8 | 42.1 |
| | LLM Call 2 | 5 | 3 | 60.0 |
| Test | LLM Call 1 | 39 | 22 | 56.4 |
| | LLM Call 2 | 5 | 4 | 80.0 |

*Table 12.* Similarity between Generated Tests & Existing Tests with Similarity Score.

| Generated Test | Test from Repo | Similarity Score |
|---|---|---|
| ```def test_call_command_with_mutually_exclusive_group(self):
  out = StringIO()
  with self.assertRaises(CommandError) as cm:
    management.call_command("my_command", shop_id=1, stdout=out)
    self.assertIn("one of the arguments --shop-id --shop is required", str(cm.exception))``` | ```def test_language_preserved(self):
  out = StringIO()
  with translation.override('fr'):
    management.call_command('dance', stdout=out)
    self.assertEqual(translation.get_language(), 'fr')``` | 0.50 |
| ```def test_ordering_with_related_field_pk(self):
  class RelatedModel(models.Model):
    pass
  class TestModel(models.Model):
    related = models.ForeignKey(RelatedModel, on_delete=models.CASCADE)
    class Meta:
      ordering = ['related_pk']
  try:
    TestModel.check()
  except ValidationError as e:
    self.fail(f"ValidationError raised: {e}")``` | ```def test_ordering_pointing_to_foreignkey_field(self):
  class Parent(models.Model):
    pass
  class Child(models.Model):
    parent = models.ForeignKey(Parent, models.CASCADE)
    class Meta:
      ordering = ('parent_id',)
  self.assertFalse(Child.check())``` | 0.55 |
| ```def test_bulk_update_return_value(self):
  for note in self.notes:
    note.note = 'test-%s' % note.id
  with self.assertNumQueries(1):
    updated_count = Note.objects.bulk_update(self.notes, ['note'])
  self.assertEqual(updated_count, len(self.notes))``` | ```def test_simple(self):
  for note in self.notes:
    note.note = 'test-%s' % note.id
  with self.assertNumQueries(1):
    Note.objects.bulk_update(self.notes, ['note'])
  self.assertCountEqual(
    Note.objects.values_list('note', flat=True),
    [cat.note for cat in self.notes]
  )``` | 0.61 |
| ```def test_composed_queries_with_values_list(self):
  ReservedName.objects.create(name='a', order=2)
  qs1 = ReservedName.objects.all()
  result1 = qs1.union(qs1).values_list('name', 'order').get()
  self.assertEqual(result1, ('a', 2))
  result2 = qs1.union(qs1).values('order').get()
  self.assertEqual(result2, (2,))``` | ```def test_union_with_values(self):
  ReservedName.objects.create(name='a', order=2)
  qs1 = ReservedName.objects.all()
  reserved_name = qs1.union(qs1).values('name', 'order', 'id').get()
  self.assertEqual(reserved_name['name'], 'a')
  self.assertEqual(reserved_name['order'], 2)
  reserved_name = qs1.union(qs1).values_list('name', 'order', 'id').get()
  self.assertEqual(reserved_name[:2], ('a', 2))``` | 0.66 |
| ```def test_type_hints_in_uml_generation(project):
  klass = project.get_module("data.clientmodule_test")["C"]
  assert hasattr(klass, "instance_attrs_type")
  type_dict = klass.instance_attrs_type
  assert len(type_dict) == 1
  keys = sorted(type_dict.keys())
  assert keys == ["a"]
  assert isinstance(type_dict["a"][0], astroid.bases.Instance), type_dict["a"]
  assert type_dict["a"][0].name == "str"``` | ```def test_instance_attrs_resolution(project):
  klass = project.get_module("data.clientmodule_test")["Specialization"]
  assert hasattr(klass, "instance_attrs_type")
  type_dict = klass.instance_attrs_type
  assert len(type_dict) == 2
  keys = sorted(type_dict.keys())
  assert keys == ["_id", "relation"]
  assert isinstance(type_dict["relation"][0], astroid.bases.Instance), type_dict[
    "relation"
  ]
  assert type_dict["relation"][0].name == "DoNothing"
  assert type_dict["_id"][0] is astroid.Uninferable``` | 0.73 |
| ```def test_classmethod_property(app):
  actual = do_autodoc(app, 'property', 'target.properties.Foo.classmethod_prop')
  assert list(actual) == [
    '',
    '.. py:property:: Foo.classmethod_prop',
    '   :module: target.properties',
    '   :type: str',
    '',
    '   Some class property.',
    '',
  ]``` | ```def test_properties(app):
  actual = do_autodoc(app, 'property', 'target.properties.Foo.prop')
  assert list(actual) == [
    '',
    '.. py:property:: Foo.prop',
    '   :module: target.properties',
    '   :type: int',
    '',
    '   docstring',
    '',
  ]``` | 0.82 |

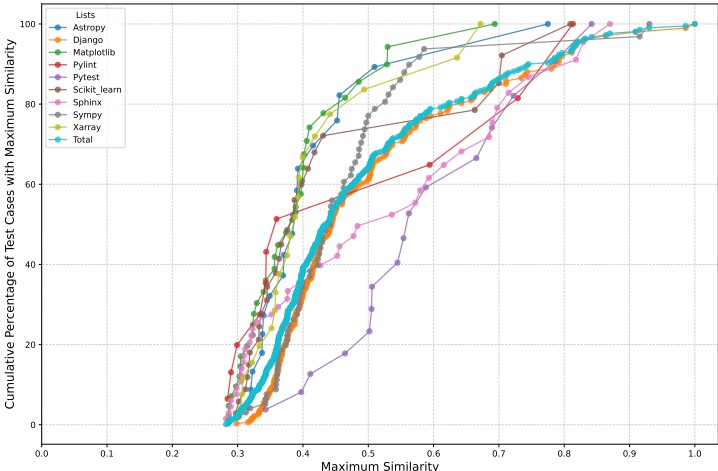

*Figure 10.* Cumulative percent of Otter generated all tests, using GPT-4o, with maximum similarity less than the similarity value shown on the x-axis.

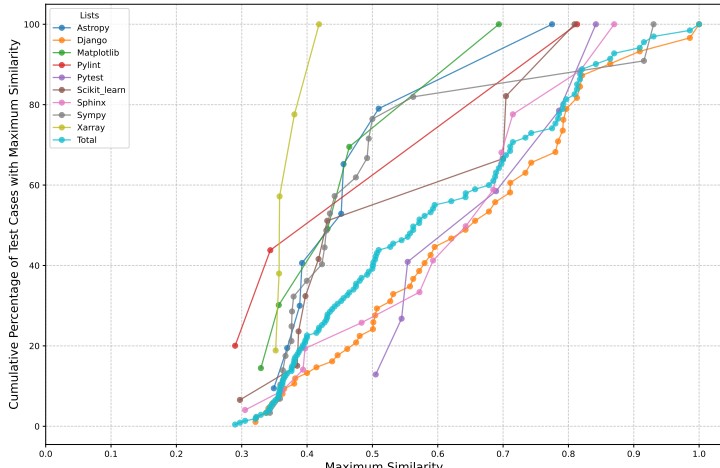

*Figure 11.* Cumulative percent of Otter generated modified tests, using GPT-4o, with maximum similarity less than the similarity value shown on the x-axis.

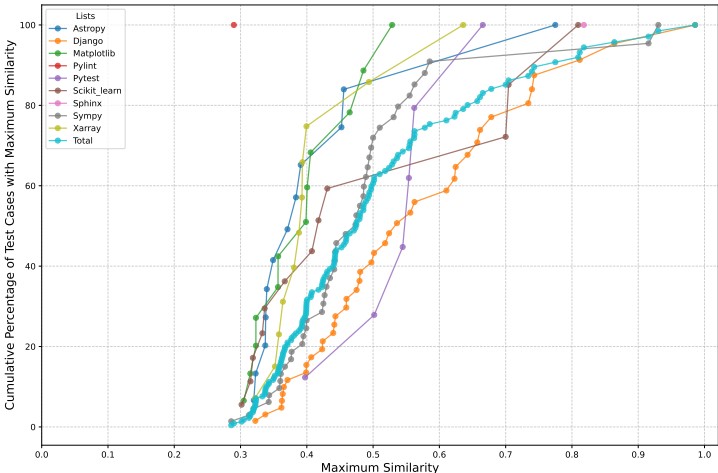

*Figure 12.* Cumulative percent of Otter generated fail-to-pass tests, using GPT-4o, with maximum similarity less than the similarity value shown on the x-axis.

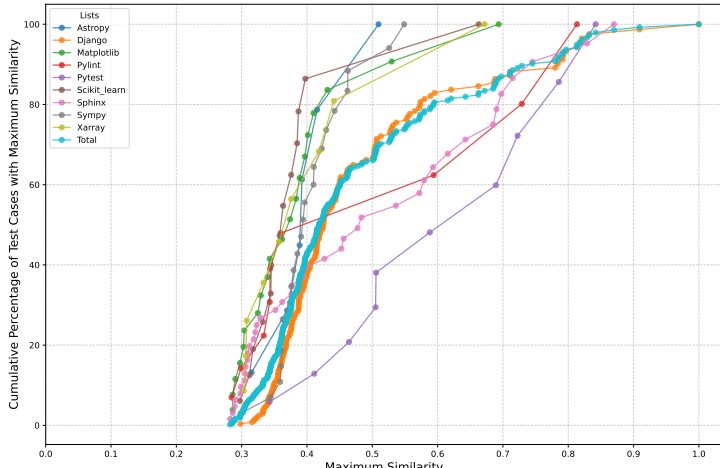

*Figure 13.* Cumulative percent of Otter generated non fail-to-pass tests, using GPT-4o, with maximum similarity less than the similarity value shown on the x-axis.

Suppose you are a very experienced developer. An issue has been created, and you need to choose the best possible file to make the changes. You will be given the following pieces of information. Please write down the file name after the "Answer" token.

   1. Repository name: $REPO_NAME

   2. Issue Description: $ISSUE_DESCRIPTION

   3. List of file(s), that will be updated or added for addressing the pull request:

$LIST_FILES

Please write down the most suitable 10 file names from the list above based on the issue description. Write one file name in each line. Do not add any number or index. Also do not add any explanation.

Answer:

*Figure 14.* Prompt (1 of 2) for focal function localizer.

Suppose you are a very experienced developer. An issue has been created, and you need to choose the best possible files and functions to make the changes. You will be given the following pieces of information. Please write down the file name and function name after the "Answer" token.

   1. Repository name: $REPO_NAME

   2. Issue Description: $ISSUE_DESCRIPTION

   3. List of file(s) and function(s), that will be updated or added for addressing the pull request:

$LIST_FILE_FUNCTION

Please write down the most suitable program files and relevant function names based on the issue description. You can choose multiple functions from different files but keep the list as short as possible.

Filename should be written between <Filename> & </Filename> tags.
Function should be written between <Function> & </Function> tags.

Each line should start with a file name and the function names written in that file. Please write down the function names from the same file in one line.

Answer:

*Figure 15.* Prompt (2 of 2) for focal function localizer.

Suppose you are a very experienced developer. An issue has been created, and you need to choose the best possible test file to write a fail-to-pass test. You will be given three pieces of information. Please write down the test file name after the "Answer" token.

   1. Repository name: $REPO_NAME

   2. Issue Description: $ISSUE_DESCRIPTION

   3. List of file(s), that will be updated or added for writing fail-to-pass test:

$LIST_FILES

Please write down the most suitable 10 test file names from the list above based on the issue description. Write one file name in each line. Do not add any number or index. Also do not add any explanation.

Answer:

*Figure 16.* Prompt (1 of 2) for test function localizer.

Suppose you are a very experienced developer. An issue has been created, and you need to choose the best possible test file to write a fail-to-pass test. You will be given three pieces of information. Please write down the file name and function name after the "Answer" token.

   1. Repository name: $REPO_NAME

   2. Issue Description: $ISSUE DESCRIPTION

   3. List of file(s), that will be updated or added for writing fail-to-pass test:

$LIST_FILE_FUNCTION

Please write down the most suitable test files and relevant function names based on the issue description. You can choose multiple test from different files but keep the list as short as possible.

Filename should be written between <Filename> & </Filename> tags.
Function should be written between <Function> & </Function> tags.

Each line should start with a file name and the function names written in that file. Please write down the function names from the same file in one line.

Answer:

*Figure 17.* Prompt (2 of 2) for test function localizer.

Suppose you are a very experienced developer. An issue has been created, and you need to write a test to ensure the issue has been resolved. You will be given the following pieces of information.

  1. Repository name: $REPO_NAME

  2. Issue Description: $ISSUE_DESCRIPTION

  3. Test file(s) and function(s), that will be updated or added for testing the pull request :

$LIST_TEST_FILE_FUNCTION

  4. Program file(s) and functions(s), that will be updated or added for addressing the issue :

$LIST_FOCAL_FILE_FUNCTION

Now make an action list to write a fail-to-pass test. You can read necessary test and focal function to finish the task. You can only perform read actions.

Read: You can read a focal function/test given a file. You can read only the current version of the focal function(s)/test(s).

Each line will have one action, file name, and focal function/test.

Action should be written between <Action> & </Action> tags
Filename should be written between <Filename> & </Filename> tags
Focal function/test should be written between <Function> & </Function> tags

Please do not repeat any specific action on the same file and focal function/test pairs. You should try to identify both focal function and test, based on the issue description. In most cases, choosing one focal function and one test is enough.

Answer:

*Figure 18.* Prompts for "make an initial plan" step.

Suppose you are a very experienced developer. An issue has been created, and you need to write a test to ensure the issue has been resolved. You were given the following pieces of information. You proposed a set of actions. We found that some of the actions were valid, while others had invalid function/test names. We will present those actions below.

  1. Valid Action(s):

$VALID_ACTION

  2. Invalid Action(s):

$INVALID_ACTION

Now, based on the action rule, you can modify the action provided with the information above (e.g., replace wrong or None function name and file name with existing functions and files). We are sharing the following information.

  1. Repository name: $REPO_NAME

  2. Issue Description: $ISSUE DESCRIPTION

  3. Test file(s) and function(s), that may be relevant to the issue:

$LIST_TEST_FILE_FUNCTION

  4. Program file(s) and functions(s), that will be updated or added for addressing the issue :

$LIST_FOCAL_FILE_FUNCTION

Action Rules to Follow:

Now make an action list to write a fail-to-pass test. You can read necessary test and program files to finish the task. You have 3 possible actions.

Read: You can read a function given a file and function name. You can read only the current version of the functions.
Write: You can write a test given a file, test function name.
Modify: You can modify a function given a file and test. Use this if you are modifying an existing test and you need to make minimal changes.

Try to guess the "Write"/"Modify" from the issue description and existing functions. If you think you cannot achieve the expected fail-to-pass function by modifying one or two lines, then write a new test for the issue. You can also propose a write if the existing test is too big and it is convenient to write a new one for the issue.

Each line will have one action, file name, and function name.

Action should be written between <Action> & </Action> tags
Filename should be written between <Filename> & </Filename> tags
Function should be written between <Function> & </Function> tags

After reading the functions mentioned in the valid action list, you may find that some of the functions or tests are irrelevant. Please drop the unnecessary read actions. You should try to identify both focal function and test, based on the issue description. In most cases, choosing one focal function and one test is enough. You are allowed to modify or write only one test. Just write down the action list only; no need to explain anything.

After writing the new action list, also write your thoughts on the action list. You have 3 options: "Satisfied", "Unsatisfied", and "Unsure".

Thought should be written between <Thought> & </Thought> tags. It should be the last line of the output, and nothing should be generated after the thought.

Answer:

*Figure 19.* Prompts for "reflect and improve the plan" step.

Suppose you are a very experienced developer. An issue has been created, and you need to write a fail-to-pass test to ensure the issue has been resolved.

After reading the issue related and existing tests, you decided to modify a test. Now, you have to write a complete test(s). We are sharing the following information.

  1. Repository name: $REPO_NAME

  2. Issue Description: $ISSUE DESCRIPTION

  3. Relevant Function:

$FUNCTION

  4. Test File imports and structure:

$FILE_FUNCTION

  5. Name of the function to be modified: $NAME

Writing Rule:

Write the complete test between the <COMPLETE_FUNC> & </COMPLETE_FUNC> tags. No need to write any explanation or add any class within the tags. Write down the fail-to-pass test. To keep the indentation intact, start writing the function on a new line using the following format:

<COMPLETE_FUNC>

# The function will be added here

</COMPLETE_FUNC>

Remember, based on the issue description the test should fail on the existing version, which may appear in 3 ("Relevant Function") and pass on the pull request. In most cases, the necessary imports are present in 4 (Test File imports and structure). Do not repeat any imports and include missing import within function body. Make sure to use the name given in 5 (Name of the function to be modified). Please write down the simplest fail-to-pass test, and discard other irrelevant fact(s) from the existing test.

Answer:

*Figure 20.* Prompt for modifying existing test.

Suppose you are a very experienced developer. An issue has been created, and you need to write a fail-to-pass test to ensure the issue has been resolved.

After reading the issue related and existing tests, you decided to write a new test function. Now, you have to write a complete function(s). We are sharing the following information.

  1. Repository name: $REPO_NAME

  2. Issue Description: $ISSUE_DESCRIPTION

  3. Relevant Function:

$FUNCTION

  4. Test File imports and structure:

$FILE_FUNCTION

  5. Name of the function to be written: $NAME

Writing Rule:

Please write the name of the prior function, after which the new function will be added. Please consider class name and indentation provided in 4 (Test File imports and structure) while proposing the name.

Prior function name should be written between <PriorFunction> & </PriorFunction> tags. No need to add any signature or parameters.

Write the complete test between the <COMPLETE_FUNC> & </COMPLETE_FUNC> tags. No need to write any explanation or add any class within the tags. Write down the fail-to-pass test. To keep the indentation intact, start writing the function on a new line using the following format:

<COMPLETE_FUNC>

# The function will be added here

</COMPLETE_FUNC>

Remember, based on the issue description the test should fail on the existing version and pass on the pull request. In most cases, the necessary imports are present in 4 (Test File imports and structure). Do not repeat any imports. Please write down the simplest fail-to-pass test.

Answer:

*Figure 21.* Prompt for writing new test.

