# OpenReview forum: "Otter: Generating Tests from Issues to Validate SWE Patches"
_ICML.cc/2025/Conference — ICML 2025 poster_

### Official Review · Reviewer_ZeyC · 2025-02-17

**Overall Recommendation:** 3

**Summary:**

This paper introduces a novel code agent specifically designed to generate test cases for real-world repositories, comprising a localizer, action planner and generator. A variant of the agent executes all but one of these components and picks the best results using some heuristics and execution feedback. The resulting agent achieves state of the art performance on the related SWT-bench and high performance on the similar benchmark introduced by the authors for the same purpose, TDD-Bench.

**Claims And Evidence:**

Several claims regarding related work are mentioned further below.
Claims regarding state-of-the-art performance appear well-founded.

**Essential References Not Discussed:**

None that I am aware of.

**Experimental Designs Or Analyses:**

While the proposed method proposes interesting components to combat likely issues with test generation, from a scientific perspective I feel there is a lack of rigor in assessing the individual component's impact (listed below in more details).

**No ablation on the localization, planner, test generator component**
In Section 4.1 a Localizer is introduced, and it is introduced that only the top 10 retrieved files are passed to the LLM. It would be interesting to see an ablation here on how the choice of less or more files influences the model performance (either qualitatively or quantitatively).
Also hallucinated file names are not dropped but replaced with minimal edit distance files. Why was this introduced and can the authors provide examples or statistics on how frequent such hallucinations are (and how useful their replacement is)?
The same holds for the planner component - the model revises its plan 5 times. Why this many times?
Finally the test generator extracts file structure and imports for the model. How exactly is the "test structure" extracted? What happens when this is not provided to the LLM?

**No analysis of relevance for various fixing steps**
The authors introduce the above mentioned hallucination replacer and an import fixer (Section 4.1 Test Generator). It would be interesting to see their impact, i.e. how many instances would fail if not for these components?

**No analysis of heterogenous prompts vs temperature**
The authors mention they use 5 heterogenous prompts at temperature 0 for their models because this proved useful in early experiments. It would be interesting to see concrete examples of how this change improves the method or some quantitative comparison of performances.

**Methods And Evaluation Criteria:**

The authors introduce fail to pass (FTP) scores and a tddScore, which is the multiplication of the FTP score with the % of changed lines covered. While the latter is an interesting score, many readers are more familiar with line coverage and may be interested in having line coverage displayed along FTP and tddScore.

**Other Comments Or Suggestions:**

Table 8 is barely readable when printed on paper, I suggest to make it slightly larger.

**Other Strengths And Weaknesses:**

The paper introduces a novel and interesting follow-up for the work of Mundler et. al., demonstrating a variety of components designed for addressing common issues during test generations.

**Questions For Authors:**

Please see my open questions in the above fields.

**Relation To Broader Scientific Literature:**

The approach is similar to the mentioned AEGIS. However, the field of repository-level test case generation is evolving and it is interesting to see different proposals for code agents for test generation (similar to code agents for code repair).


In section 4.1 "Test generator", the authors claim that Mundler et. al. attempt to generate diffs directly, however telling from the corresponding paper, Mundler et.al. actually introduce a custom diff format in their zero-shot setups that mainly asks the model to generate complete functions and where to insert them, similar to the method that the authors propose. The agents do not generate diffs either but are equipped with different editing tools. I suggest the authors adapt these claims accordingly or clarify how they consider their method to differentiate.

In Section 6.4, the authors moreover claim that "two of [Mundler et al.'s] approaches use a golden patch in the prompt". Could the authors clarify which methods these are?

For Otter++, an execution feedback oracle is used, concretely, the best model generation is picked based on execution logs in the final setting. While this is similar to the LIBRO setting in Mundler et. al., it is a clear advantage over their analysed code agents which do not have access to a correctly set-up environment and the test results. The authors may consider mentioning this, since it may be a contributing factor to their superior performance.

**Theoretical Claims:**

No theoretical claims made/checked.

---

> ### Author Rebuttal · Authors · 2025-03-31
>
> a) Reporting Coverage: We will add a coverage column to Table 1.
>
> |     Model     |  Approach | Coverage |
> |-|-|-|
> | Mistral | Zero-shot |     60.6 |
> |               | Otter     |     70.5 |
> |               | Otter++   |     70.4 |
> | GPT-4o         | Zero-shot |     60.0 |
> |               | Otter     |     70.6 |
> |               | Otter++   |     71.5 |
>
>
>
> b) Ablation: We have conducted a detailed analysis of Otter tests in response to Reviewer G2zQ's comments (see- Analyze Otter-generated tests) and have modified our ablation study to further break down self-reflective action planning (Reviewer BFCj- Clarification on Ablation).
>
> File Retrieval: We evaluated the effect of the number of localized files on the results and found it to be minor:
>
> Focal (#of files, accuracy) = {(5, 82.4%), (10, 82.4%), (15, 83.3%)}
>
> Test (#of files, accuracy) = {(5, 69.9%), (10, 70.6%), (15, 71.9%)}
>
> Hallucination: The localizer makes two LLM calls during which hallucinations can occur. In the first call, we drop hallucinated file names. In the second call, we replace hallucinated file names with existing ones to ensure our pipeline functions correctly.
>
> | Type |  LLM Call  | #of Hallucinations Happen | #of fail-to-pass  | in %|
> |-|-|-|-|-|
> | Focal      | 1 |                       19 |                 8 |              42.1 |
> |            |  2 |                        5 |                 3 |              60.0 |
> | Test       |  1 |                       39 |                22 |              56.4 |
> |            |  2 |                        5 |                 4 |              80.0 |
>
> The results show that our proposed technique helps Otter achieve better performance. For example, if we didn’t replace the hallucinated file name in the 2nd call of test localization (the last row), we would have lost 5 samples because our pipeline would have exited with an error. Replacing the hallucinated name generated 4 fail-to-pass tests for those samples.
>
> Plan revision up to 5 times: The model does not always revise the plan 5 times. If the model is satisfied before making 5 calls, it does not make further revisions. Figure 4 shows that the LLM is usually satisfied within 2 calls. Our approach could go beyond 5 calls, but considering the cost and time, we limit the number of plan revisions to 5.
>
> Test File Structure: It contains lines where classes and functions are defined. We build the test file structure by omitting lines that do not define a new class or function. Note that 72.8% tests are new, and test file structure is necessary for those tests. LLMs are unable to indicate the position of new tests without it.
>
> c) Analysis for various fixing steps: We have already added the results in Table 2 for the import fixer. We will lose 11 and 4 instances with GPT-4o and Mistral-large, respectively. We address the hallucination above.
>
> d) Heterogenous Prompts vs Temperature: We make multiple LLM calls in different stages and multi-sampling in each stage will exponentially increase the test counts. To compare heterogenous prompting with temperature, we generated 5 samples at high temperature (1.0) in our last LLM call in Test Generator phase of Otter (our best solution). Though the average number of fail-to-pass test goes up (117.8 vs 116.6), the Top-5 and Top-1 (using Otter++ ranker) results remain lower (Top-5: 173 vs 197 and Top-1: 146 vs 166). That means heterogenous prompting boosts up the overall ensemble performance if we compare at the same number of samples.
>
> e) Diff format: The custom-diff has similarities to our approach. However, the model needs to perform additional tasks such as writing the file name, change type, and line number in response to one LLM call. Additionally, to instruct the model to generate a specific format, the authors had to include an example that is not relevant to the issue itself. In our generation step, the model only needs to generate the test (and prior function name for positioning new tests).
>
> f) Usage of patch in Mundler et al.'s: Thank you for pointing this out. Figure 12 of the SWT-Bench paper presents the prompt that was used for ZeroShotPlus and LIBRO. Line 7 indicates that the “proposed patch” has been used to generate the test. We mistakenly thought this refers to the golden patch, which we now realize is not the case. We will fix this statement in the paper.
>
> g) Execution feedback: Otter++ uses execution logs on the current code base and gains an advantage in the final selection stage. We will mention the use of feedback as a contributing factor to the superior performance of Otter++. Note that Otter does not use any execution logs, yet it still performs significantly better than LIBRO (25.4% vs 15.2% in Table 3).
>
> h) Supplementary Material: We will improve the Otter repository as suggested. We removed the license following the ICML author instructions. We will address this in our open-source version. V5 is our internal name used for Otter; we apologize for the confusion it caused.

---

> > ### Comment · Reviewer_ZeyC · 2025-04-02
> >
> > I thank the authors for their extensive additional experiments and corrections.
> >
> > > Though the average number of fail-to-pass test goes up (117.8 vs 116.6), the Top-5 and Top-1 (using Otter++ ranker) results remain lower (Top-5: 173 vs 197 and Top-1: 146 vs 166).
> >
> > Can you please elaborate this a bit further? Does this not simply imply that the ranker is not very good? It seems high temperature/heterogenous prompts does not make a significant difference before the ranker. I think heterogenous prompts is a good idea to increase sample diversity but temperature should not be so simply discarded without convincing evidence of its inferiority. This should be discussed more in the next revision.
> >
> > I think the paper is potentially interesting to the test generation community when the discussed changes are incorporated and accordingly raised my score.

---

> > > ### Author Response · Authors · 2025-04-02
> > >
> > > Thank you for your comment! We agree that the question of high temperature vs. heterogeneous prompts deserves more discussion in the revised paper. We will make sure the revised paper includes that. We expanded the experiment from our previous author response:
> > >
> > > | Setting                                   | Individual Performance                   | Max | Average | Top-1 | Oracle |
> > > |-------------------------------------------|------------------------------------------|-----|---------|-------|--------|
> > > | 5 heterogeneous prompts (Otter++)         | 141,110,115,107,110                      | 141 | 116.6   | 166   | 197    |
> > > | 5 high-temperature samples (temp=1.0)     | 111,118,126,121,113                      | 126 | 117.8   | 146   | 173    |
> > > | 5 = 1 greedy + 4 high-temperature samples | 141, 111,118,126,121                     | 141 | 123.4   | 152   | 183    |
> > > | 10 = 5 heterogeneous + 5 high-temperature | 141,110,115,107,110, 111,118,126,121,113 | 141 | 117.2   | 168   | 218    |
> > >
> > > Here, the Top-1 column shows "fail-to-pass @ 1" based on our actual ranker, whereas the Oracle column shows "fail-to-pass @ N", i.e., the result if we had a perfect ranker.  Heterogeneous prompts yield better results than high-temperature samples in both the Top-1 column and the Oracle column. This is evidence that the difference indeed comes from the diversity of heterogeneous prompts and is not ranker specific. The difference between the Top-1 and the Oracle column is almost the same in the first three rows (i.e., all settings with 5 samples). That said, there is still room for improvement in the ranker, as illustrated by the last row. In the last row, an oracle that could reliably choose among all 10 samples would perform amazingly well, motivating further work on better rankers.

---

### Official Review · Reviewer_b7T8 · 2025-03-12

**Overall Recommendation:** 2

**Summary:**

This paper introduces Otter, a system for generating tests from issue descriptions to validate software engineering (SWE) patches. Unlike prior work that focuses on generating tests for existing code, Otter addresses the scenario where a code patch does not yet exist, supporting test-driven development (TDD) and validation of SWE agents. The system takes an issue description and original code as input, and generates tests that should fail on the old code but pass on the new code after the issue is resolved.

**Claims And Evidence:**

The paper's main claims are supported by evidence.

**Essential References Not Discussed:**

As mentioned above, the following references are essential to understanding the context but are not cited:

```
[1] Wang X, Gao P, Meng X, et al. (2024) "AEGIS: An Agent-based Framework for General Bug Reproduction from Issue Descriptions" - This work directly addresses bug reproduction from issue descriptions using a multi-agent framework, which is highly relevant to Otter's goals.

[2] Lin Y, Ma Y, Cao R, et al. (2024) "LLMs as Continuous Learners: Improving the Reproduction of Defective Code in Software Issues" - This paper explores similar problem space of reproducing issues from descriptions.

[3] Ehrlich R, Brown B, Juravsky J, et al. (2025) "CodeMonkeys: Scaling Test-Time Compute for Software Engineering" - This work presents related approaches for test generation that would provide important context.
```

**Experimental Designs Or Analyses:**

The experimental design is generally sound:

- The comparison with baselines uses the same dataset and evaluation metrics.
- The ablation studies help understand component contributions.
- The data contamination analysis addresses potential concerns about model memorization.

**Methods And Evaluation Criteria:**

The methods and evaluation criteria are generally appropriate for the problem:
- The TDD-Bench-Verified benchmark is a reasonable extension of SWE-bench Verified with additional quality filters.
- The tddScore metric combines fail-to-pass behavior with code coverage, which is appropriate for evaluating test quality.
- The comparison with existing approaches (SWE-Agent+, LIBRO, etc.) provides context for Otter's performance.

However, there are some limitations:

- The paper would benefit from more qualitative analysis of the generated tests to understand their characteristics beyond just pass/fail rates.
- The evaluation focuses primarily on Python repositories, which may limit generalizability.

**Other Comments Or Suggestions:**

- The paper would benefit from more examples of generated tests in the main text.
- The paper should address how Otter might be integrated into existing developer workflows.

**Other Strengths And Weaknesses:**

Strengths:
- The paper addresses an important practical problem in software engineering - generating tests from issue descriptions before patches exist.
- The comprehensive evaluation across multiple models and comparison with existing approaches provides a clear picture of Otter's capabilities.
- The introduction of TDD-Bench-Verified is a valuable contribution to the research community.

Weaknesses:

- **Limited novelty**: The primary novelty is in the combination of existing techniques rather than fundamentally new approaches.
- **Incomplete ablation studies**: While Table 2 shows component contributions, more detailed analysis of the novel components (especially the self-reflective planning) would strengthen the paper.
- **Limited qualitative analysis**: Beyond pass/fail rates, deeper analysis of test characteristics would be valuable.
- **Missing comparisons with concurrent work**: The paper doesn't compare with or discuss several highly relevant recent papers in the same problem space.

**Questions For Authors:**

See about "Other Strengths And Weaknesses".

**Relation To Broader Scientific Literature:**

The paper positions itself well within the literature on test generation and SWE agents. It builds upon:
- SWE-bench and SWE-bench Verified for issue resolution benchmarking
- Recent work on LLM-based SWE agents (SWE-Agent, CodeR, SpecRover)
- Test generation systems like LIBRO

**Theoretical Claims:**

The paper doesn't make significant theoretical claims requiring formal proofs.

---

> ### Author Rebuttal · Authors · 2025-03-31
>
> a) Limited novelty: Our approach introduces self-reflective action planning for generating bug-reproduction tests, which is a novel application in this problem setting. Our approach also incorporates heterogeneous prompting for multi-sampling, which is shown to be better than multi-sampling at higher temperatures. Although applying static analysis for repairing name-related errors, prompt design, and rule-based checking are not new, we have adapted them to work well in our problem setting. Together these features enable our approach to achieve significantly better results than the techniques presented by Mundler et al.
>
> b) Incomplete ablation studies: We have modified and restructured the ablation study to place more focus on self-reflective action planning (See reviewer BFCj “Clarification on Ablation”). We have added the last two rows in the following table (Table 2 in the paper): Row 2 shows the number of fail-to-pass tests obtained if plan refinement is disabled; Row 3 shows the results obtained if we do not provide the model with valid and invalid action lists.
>
> |  Model |    Component   |                 Approach                 | # of fail-to-pass | tddScore | Change in tddScore% |
> |:-:|:-:|:-:|:-:|:-:|:-:|
> |   GPT-4o  | Action Planner | Otter - Action Planning (complete)       |               110 |     23.6 |               -19.7 |
> |        |                | Otter - Plan Refinement (Just 1 attempt) |               130 |     27.5 |                -6.5 |
> |        |                | Otter - Action Validation               |               120 |     25.7 |               -12.6 |
>
> In Figure 4 of the paper, we present the number of turns taken in the “reflect and improve the plan” step. In the following table, we show the total number of actions proposed by the model in the final stage and how the performance changes with it. We find that the number of actions tends to be around 1-3, with no clear correlation between the number of actions and whether the test ultimately satisfies the fail-to-pass criterion.
>
> | #of Action | #of Sample | #of fail-to-pass  | in % |
> |-|-|-|-|
> | 1          | 18         | 7                 | 38.9 |
> | 2          | 157        | 49                | 31.2 |
> | 3          | 258        | 78                | 30.2 |
> | 4          | 15         | 7                 | 46.7 |
> | 5          | 1          | 0                 | 0    |
>
> c) Limited qualitative analysis: We have added a detailed analysis of the Otter-generated test in response to Reviewer G2zQ (Analyze Otter-generated tests). We discuss how performance changes with mislocalization, the test status on the current codebase, and the type of change (new/modified).
>
> d) Missing comparison with concurrent work: Thank you for pointing out the references to Aegis, EvoCoder, and CodeMonkeys. All three of them are highly relevant to our work, as they involve generating reproduction tests from issues. We will make sure to cite and discuss them.
>
> [1] Aegis: Our related-work section already cites and discusses this paper.
>
> [2] EvoCoder: This paper proposes the interesting idea of using experiences from prior issues to help with the latest issue at hand, which is complementary to Otter. Unlike our work, the generated tests are not integrated with the existing CI pipeline. Furthermore, the experiments do not use execution-based metrics, making it hard to compare empirically.
>
> [3] CodeMonkeys: This paper leads with inference scaling and patch generation. It appeared on arXiv on Jan 24, 2025, and we did not realize its test-generation contributions until after we had submitted our work to ICML on Jan 30. CodeMonkeys iteratively and jointly improves a patch with a test, which is interesting and complementary to Otter. Unlike our work, their experiments do not evaluate the fail-to-pass rate nor coverage of generated tests.
>
>
> e) Example & Integrating Otter: Thank you for your suggestions. We will add more examples in the main text. We envision two ways of integrating Otter into existing developer workflows. First, Otter can run on a new issue to propose a test, which the stakeholders can use to clarify requirements for the desired behavior after issue resolution and, following that, a developer can use the test for test-driven development to resolve the issue. Second, Otter can be paired with a patch-generation solution (a "SWE agent") to create a PR that includes both a patch and a test.

---

> > ### Comment · Reviewer_b7T8 · 2025-04-02
> >
> > Thank you for the author's response. I appreciate the clarification regarding the novelty aspects of the paper. While these adaptations (self-reflective action planning for bug-reproduction tests and the heterogeneous prompting for multi-sampling) show thoughtful engineering and integration of techniques for this specific problem domain, I still believe the primary contribution lies in the effective combination of existing approaches rather than fundamentally novel methods. The significant performance improvements over Mundler et al. are certainly valuable, but my original assessment of the novelty limitation remains. Therefore, I will maintain my original score for the paper.

---

> > > ### Author Response · Authors · 2025-04-03
> > >
> > > Thank you for your comments, which will greatly help us to improve our work. We are happy to provide clarifications if you have any further concerns.

---

### Official Review · Reviewer_BFCj · 2025-03-14

**Overall Recommendation:** 4

**Summary:**

The study describes a system to produce tests from code issues by leveraging LLMs with rule-based analysis and multi-step process with feedback. In addition, the study provides an associated benchmark derived from the SWE-bench Verified benchmark for evaluating tests generated from issues.

**Claims And Evidence:**

Yes.

**Essential References Not Discussed:**

I do not identify and major omissions in the references.

**Experimental Designs Or Analyses:**

I read the descriptions of the experiments.

**Methods And Evaluation Criteria:**

Yes.

**Other Comments Or Suggestions:**

Text and numbers in many of the figures are too small to read (Fig 4-8)

In the future, please enable the submission status on the template. It makes it easier to review.

The textual reference to Mundler et al. should be standardized: Mundler et al. 2024

The use of “cov” in the equations is confusing since usual “cov” refers to covariance.

Table 2 is confusing but one that I am interested in. It looks like this is an ablation test focusing on the “fixing import” and “import at generation”  steps for Otter. It’s hard to match where those are in Figure 1. It would be interesting (and increase the impact of the work) to explicitly split out between the LLM, LLM w/ feedback, and rule-based components. The other entries look like they are ablation tests for Otter++ and not Otter.

**Other Strengths And Weaknesses:**

The study is fairly comprehensive and clear– describing the methodology, associated benchmark, and assessments of both (including cost).
However, I think that one area of potential improvement is clarifying the ablation study to capture the effect of the novel components of the workflow (the rule based and reflective components) clearly.

**Questions For Authors:**

Can you provide further detail into what the issue is with coverage on sympy and what you did for manual validation?

When identifying “local files” how do you handle inheritance? E.g. if a class inherits a method from a parent class do you include both files?

**Relation To Broader Scientific Literature:**

Test generation for issues is generally relevant. The workflow described combining LLMs with rule-based steps is interesting but could be presented better.

**Theoretical Claims:**

N/A

---

> ### Author Rebuttal · Authors · 2025-03-31
>
> a) Issue with Sympy Coverage: We use the Python coverage package in the TDD-Bench-Verified benchmark, which works well with Pytest. However, the Sympy project does not use Pytest for testing. Additionally, the coverage package may fail due to dependency issues. In Sympy, we were able to generate the coverage report but found that many fail-to-pass tests had zero coverage. We then manually checked several of the fail-to-pass tests and found inconsistencies in the coverage report. There were lines that were executed but not covered by the package. Therefore, we decided to exclude coverage from the TDD-Score metric for Sympy.
>
> b) Handling Inheritance: The Otter pipeline does not explicitly handle inheritance, but the localizer can include functions from the parent class if it deems them relevant to the issue description. In several instances, the localizer selects multiple functions from different files. Also, the localizer does not consider library functions because it only examines user-defined functions from the current repository.
>
> c) Clarification on Ablation: We apologize for the confusion with respect to Table 2, which presents the ablation study of Otter, not Otter++. Figure 2 offers an overview of Otter, which comprises three major components: localizers, a self-reflective action planner, and a test generator. Please see the modified ablation study for the GPT-4o model (the Mistral-large version will be added in the final release) in the table below.
>
> In the first group, we present the ablation on the action planner. The first row shows the results when the action planner is removed. The following two rows demonstrate the outcomes when action planning is retained, but action refinement and action validation input to the LLM are disabled. Note that without self-reflective action planning, we cannot refine our localizations. Therefore, we must present all the functions collected during focal and test localizations.
>
> In the second group, we present the ablation on the localizer. The table shows the results of omitting either or both localizations. We did not run self-reflection on partial or no localization because, without both localizations, self-reflective action planning does not add any value.
> The last two entries, “fixing import” and “imports at generation”, belong to our third component, the test generator. These are the only two rule-based entries.
>
> |  Model |    Component   |                 Approach                 | # of fail-to-pass | tddScore | Change in tddScore% |
> |:-:|:-:|:-:|:-:|:-:|:-:|
> | GPT-4o | NA             | Otter                                    |               141 |     29.4 |                  NA |
> |        | Action Planner | Otter - Action Planning (complete)       |               110 |     23.6 |               -19.7 |
> |        |                | Otter - Plan Refinement (Just 1 attempt) |               130 |     27.5 |                -6.5 |
> |        |                | Otter - Action Validation               |               120 |     25.7 |               -12.6 |
> |        | Localizers     | Otter - Focal Localization              |               115 |     25.2 |               -14.3 |
> |        |                | Otter - Test Localization               |               107 |     24.2 |               -17.7 |
> |        |                | Other - Focal & Test Localization      |               110 |     24.2 |               -17.7 |
> |        | Test Generator | Otter - Fixing Import                    |               128 |     26.7 |                -9.2 |
> |        |                | Otter - Imports at Generation            |               130 |     26.7 |                -9.2 |
>
> To clarify, Otter++ uses heterogeneous prompting and a rule-based ranker on execution logs. For heterogeneous prompting, we used the same prompts used for the ablation on Otter. Therefore, the execution log is not used by Otter.

---

### Official Review · Reviewer_G2zQ · 2025-03-15

**Overall Recommendation:** 3

**Summary:**

The paper introduces Otter, an LLM-based solution that generates unit tests directly from issue descriptions before a code patch is written. This approach supports test-driven development scenario and reduces LLMs overfitting of unit tests to the specific focal code. Otter employs a self-reflective action planner to iteratively decide which code to read and how to construct meaningful tests. Additionally, Otter applies rule-based post-processing to fix missing imports and validate test correctness. To evaluate its effectiveness, the paper introduces TDD-Bench-Verified, a benchmark derived from SWE-Bench Verified, which measures test quality based on fail-to-pass rates.

Empirical results show that Otter++ (an ensemble version of Otter) achieves a 37.0% fail-to-pass rate, outperforming previous methods like SWE-Agent+ (19.2%). Additionally, incorporating Otter-generated tests into automated patch validation improves precision from 60.8% to 91.9%, helping filter out incorrect fixes. The approach remains cost-effective, with test generation costing less than $0.10 per issue when using GPT-4o.

**Claims And Evidence:**

The claims are generally well supported by the evaluation, although the cost-effectiveness claim only takes into account the GPU inference costs but not the infrastructure for executing tests.

**Essential References Not Discussed:**

"CodeT: Code Generation with Generated Tests", B. Chen et. al

**Experimental Designs Or Analyses:**

The Otter paper uses a structured ablation study to isolate the impact of key components, such as self-reflective action planning, focal/test localization, and import fixing. Removing action planning reduces fail-to-pass rates by 14–20%, while eliminating test localization leads to a steeper drop (up to 36%), confirming that each step significantly improves test quality. The study also shows that test localization contributes more than focal localization, suggesting that retrieving relevant tests is more critical for fail-to-pass success. While these results validate Otter’s design, a deeper breakdown of failure cases (e.g., incorrect test logic vs. mislocalization) would further clarify its contributions. In addition, it would be good to include alternative localization approaches -- for instance, using analysis of import statements, or based line test coverage (dynamic contexts).

The paper also compares Otter against prior test-generation models on SWT-Bench Lite, showing that Otter++ (29.0%) outperforms SWE-Agent+ (19.2%) and Libro (15.2%), highlighting the benefits of self-reflection and execution-guided filtering over static LLM outputs. By evaluating Otter across TDD-Bench-Verified and SWT-Bench Lite, the study ensures its findings generalize beyond a single dataset. However, statistical significance testing is could also be conducted, making it more clear whether performance gains consistently hold across different types of issues. In addition, a comparison to other patch filtering techniques like those introduced in CodeT paper could be helpful.

**Methods And Evaluation Criteria:**

The paper evaluates Otter’s test generation quality using fail-to-pass rates and TDDScore, both measured on the TDD-Bench-Verified dataset. The fail-to-pass rate directly assesses whether generated tests meaningfully capture the reported issue by failing on the buggy code and passing after the fix, with Otter achieving 31.4% (Otter) and 37.0% (Otter++) -- outperforming prior methods like SWE-Agent+ (19.2%). The TDDScore extends this evaluation by incorporating test adequacy, measuring whether the generated tests effectively cover relevant code changes. This metric provides a more granular evaluation of test quality, ensuring that the generated tests are both correct and comprehensive.

For patch filtering, the paper measures how well Otter-generated tests improve precision and recall in validating patches using the SWE-Bench Verified dataset. The results show that filtering patches with Otter-generated tests increases precision from 60.8% to 91.9%, though recall drops to 33%, indicating a trade-off between accuracy and completeness. These results suggest that Otter-generated tests effectively eliminate incorrect patches while preserving some valid ones.

**Other Comments Or Suggestions:**

None

**Other Strengths And Weaknesses:**

Strengths:
Otter’s focus on test generation from issues aligns with real-world SWE agentic workflows, making it highly applicable for automated debugging and test-driven development in software engineering.

The introduction of TDD-Bench-Verified also provides a structured benchmark for evaluating fail-to-pass test generation, ensuring practical impact.

Weaknesses
The evaluation effectively quantifies success rates, but it does not deeply analyze why some generated tests fail (e.g., mislocalization, incorrect assertions, gaps in execution filtering, or patch applicability errors). A more detailed error categorization would provide stronger insights into where Otter struggles and how it can be further improved.

**Questions For Authors:**

The paper includes an ablation study evaluating the impact of test and focal localization on fail-to-pass rates, but could you provide further analysis on alternative localization strategies (e.g. based on imports or dynamic coverage context driven localization)?

While the evaluation measures fail-to-pass success rates, it does not deeply analyze cases where Otter-generated tests fail to apply to patches. Have you experimented with alternative diff formats, can you break down these metrics for addition-only, patches, etc?
A breakdown of these cases would help assess the robustness of Otter’s generated tests in practical debugging workflows.

The study demonstrates that Otter-generated tests improve patch filtering, but have you compared Otter’s filtering mechanism to other patch ranking algorithms, such as those used in CodeT?

How does Otter++ scale when increasing the number of generated tests per issue or expanding the number of candidate patches evaluated?

**Relation To Broader Scientific Literature:**

The Otter paper builds upon and extends prior research in LLM-driven test generation, automated software debugging, and self-correction. The techniques discussed in the paper are instrumental to building TDD-style SWE agents, where tests are generated upfront to guide issue resolution. This aligns closely with recent work in SWE-Agent, which explores GH issue resolution, SWT-Bench, which benchmarks test generation from issues, and CodeT, which to the best of my knowledge is the first to introduce the concept of test generation conditioned on the natural language problem description rather than focal code.

**Theoretical Claims:**

The Otter paper relies on empirical validation of its novel test generation approach, there are no theoretical proofs to verify.

---

> ### Author Rebuttal · Authors · 2025-03-31
>
> a) Alternative Localization Strategies: Thank you for your suggestion. To further assess the impact of localization, we re-ran Otter using oracle golden localizations for both focal and test functions. We did not observe any significant performance gain (32.5% vs 31.4% with GPT-4o) in the fail-to-pass rate. This indicates that Otter's localization is already good enough. Perfection is not required in this phase; for instance, it is fine to add a test to a different file than the golden test.
>
> b) Analyze Otter-generated tests: As per your suggestion, we have analyzed the Otter-generated tests using GPT-4o model from different perspectives, as outlined in the following table. We will include additional analysis in the final version of the paper.
> |      Perspective     |    Category   | #of Sample | #of fail-to-pass  | in %|
> |:-:|:-:|:-:|:-:|:-:|
> | Type of Test         | PatchExisting |         122 |                50 |                   41.0 |
> |                      | AdditionOnly  |         327 |                91 |                   27.8 |
> | Test on old Codebase | Pass          |          69 |                 0 |                      0 |
> |                      | AssertionFail |         170 |                97 |                   57.1 |
> |                      | Fail          |         101 |                40 |                   39.6 |
> |                      | Error         |         109 |                 4 |                    3.7 |
> | Focal Localization   | Correct       |         370 |               118 |                   31.9 |
> |                      | Wrong         |          79 |                23 |                   29.1 |
> | Test Localization    | Correct       |         317 |               117 |                   36.9 |
> |                      | Wrong         |         132 |                24 |                   18.2 |
>
> Modified/New: We see that the fail-to-pass rate is higher when patching an existing test than when adding a new test, which is expected because the model can have better context from the existing test. Writing a new test is inherently more difficult than modifying an existing test.
>
> Test on old codebase: It is expected that the test will fail on the old codebase. However, we found 69 samples in Otter where the test passed on the old codebase. Also, our analysis shows that tests with assertion failures have a higher success rate (57.1% fail-to-pass rate) compared to other groups.
>
> Focal Localization: We did not see much impact of focal localization on the performance, with 31.9% and 29.1% for correct and incorrect localization, respectively.
>
> Test Localization: Test localization has a significant impact on the performance (36.9% vs 18.2% for correct and incorrect localization). In our ablation study, we also found that test localization is more important than focal localization.
>
> Alternative diff: We did not explore alternative diff formats. Mundler et al. compared two diff formats in their paper and concluded that custom diff outperforms unified diff. Our approach shares some similarities with the custom diff format, as explained in response to reviewer ZeyC (Diff format). In brief, custom diff provides the file name, type of change (new/modified), position of the edit, and the complete class or function in response to a single call. We extract similar information from the LLM using multiple calls, while keeping the total number of calls low, with specialized prompts. This approach resulted in significantly better performance than those using different diff formats.
>
> c) CodeT Ranking: CodeT is not directly applicable to our current setup because we have multiple tests but not the solution. However, apart from filtering SWE-patches, we could use our tests to choose the best SWE-patch, which would be a good application of the Otter-generated tests. Following your suggestion, we have tried CodeT ranking on candidates from the top 3 leaderboard systems on SWE-Bench-Verified and observed 2% improvement. Note that some of these leaderboard solutions have already been through good rankers and used superior models. Improving upon these samples using a ranker may be difficult. We will add a discussion on CodeT ranking in the paper.
>
> d) Scaling of Otter++: Otter++ uses heterogeneous prompting and execution logs to select the best solution. Therefore, we cannot significantly increase the number of samples. Otter++ scaled well with samples up to 5, giving 0.5%-2.9% improvement.
>
> | #of  Candidate | #of Fail-to-pass  | In % |
> |-|-|-|
> |              1 |                    141 | 31.4 |
> |              2 |                    154 | 34.3 |
> |              3 |                    157 | 35.0 |
> |              4 |                    164 | 36.5 |
> |              5 |                    166 | 37.0 |
>
> We conducted some experiments with temperature (see Reviewer ZeyC -Heterogeneous Prompts vs Temperature) to generate more samples, but we did not find it beneficial.

---

### Decision · Program_Chairs · 2025-05-01

**Decision:**

Accept (poster)

**Comment:**

This paper introduces Otter, a system leveraging large language models combined with rule-based analysis and self-reflective planning to generate software tests directly from issue descriptions, aiming to support test-driven development and validate automated code patches. Reviewers broadly acknowledged the practical relevance of this problem setting, the system's strong empirical performance compared to state-of-the-art methods like SWE-Agent+ (G2zQ, ZeyC), and the value of the introduced TDD-Bench-Verified benchmark (G2zQ, BFCj). However, concerns were raised regarding the degree of methodological novelty (b7T8), the completeness of the initial ablation studies (G2zQ, BFCj, b7T8, ZeyC), the lack of deep qualitative analysis of failure modes (G2zQ, b7T8), and missing comparisons or discussions of some related work (G2zQ, b7T8).

The authors provided a detailed rebuttal with significant new experimental results and analyses. They presented expanded ablation studies clarifying the impact of individual components like localization and the novel self-reflective action planner, addressing points raised by multiple reviewers (G2zQ, BFCj, b7T8, ZeyC). They also included a qualitative breakdown of generated test outcomes and failure reasons (addressing G2zQ, b7T8) and committed to incorporating discussions of the related work highlighted by reviewers (addressing G2zQ, b7T8). These additions were well-received, notably leading reviewer ZeyC to increase their score, indicating that the clarifications sufficiently addressed their concerns about component contributions and analysis rigor.

While reviewer b7T8 maintained their assessment regarding limited fundamental novelty, acknowledging the work primarily as effective engineering, the demonstrated state-of-the-art results on a challenging and practically relevant software engineering task carry significant weight. Considering the strong empirical contribution, the introduction of a useful benchmark, the system's relevance to the growing field of AI for software engineering, and the positive reception of the rebuttal by several reviewers, the paper represents a solid contribution that meets the acceptance criteria for ICML.